# Enhancing Fairness in Unsupervised Graph Anomaly Detection through Disentanglement

**Wenjing Chang**                                                *changwenjing@cnic.cn*
*Computer Network Information Center, Chinese Academy of Sciences*
*University of Chinese Academy of Sciences*

**Kay Liu**                                                      *zliu234@uic.edu*
*University of Illinois Chicago*

**Philip S. Yu**                                                 *psyu@uic.edu*
*University of Illinois Chicago*

**Jianjun Yu**                                                   *yujj@cnic.ac.cn*
*Computer Network Information Center, Chinese Academy of Sciences*

**Reviewed on OpenReview:** *https://openreview.net/forum?id=5zRs34Ls3C*

## Abstract

Graph anomaly detection (GAD) is becoming increasingly crucial in various applications, ranging from financial fraud detection to fake news detection. However, current GAD methods largely overlook the fairness problem, which might result in discriminatory decisions skewed toward certain demographic groups defined on sensitive attributes (e.g., gender). This greatly limits the applicability of these methods in real-world scenarios in light of societal and ethical restrictions. To address this critical gap, we make the first attempt to integrate fairness with utility in GAD decision-making. Specifically, we devise a novel **D**is**E**ntangle-based **F**airn**E**ss-aware a**N**omaly **D**etection framework on the attributed graph, named **DEFEND**. DEFEND first introduces disentanglement in GNNs to capture informative yet sensitive-irrelevant node representations, effectively reducing bias inherent in graph representation learning. Besides, to alleviate discriminatory bias in evaluating anomalies, DEFEND adopts a reconstruction-based method, which concentrates solely on node attributes and avoids incorporating biased graph topology. Additionally, given the inherent association between sensitive-relevant and -irrelevant attributes, DEFEND further constrains the correlation between the reconstruction error and predicted sensitive attributes. Empirical evaluations on real-world datasets reveal that DEFEND performs effectively in GAD and significantly enhances fairness compared to state-of-the-art baselines. Our code is available at `https://github.com/AhaChang/DEFEND`.

## 1 Introduction

Graph Anomaly Detection (GAD), which aims to identify nodes that deviate significantly from the majority of nodes, has attracted wide attention in various domains, including fraudster detection in financial networks (Zhang et al., 2022; Huang et al., 2022) and spammer detection in social networks (Li et al., 2019; Wu et al., 2020). The advancement of Graph Neural Networks (GNNs) (Kipf & Welling, 2016; Hamilton et al., 2017; Veličković et al., 2017) has significantly enhanced the ability of GNN-based GAD methods to accurately identify anomalies (Ding et al., 2019; Chai et al., 2022; Kim et al., 2023; He et al., 2023). However, a recent study (Neo et al., 2024) reveal a concerning trend: current GAD methods exhibit substantial bias in decision.

Given the wide-ranging applications of GAD, particularly within high-stakes domains, the fairness problem cannot be overlooked. Unfair decisions that skew toward certain demographic groups associated with sen-

sitive attributes (e.g., gender, religion, ethnicity, etc.) might cause profound societal and ethical concerns. For example, in the realm of social networks (e.g., Reddit and Twitter), anomalous users (e.g., spreading misinformation or engaging in fake account interactions) might undergo strict investigation and even permanent account suspension. In such scenarios, biased decisions could result in unfairly focusing on certain groups while inadvertently neglecting others. This undermines the effectiveness and reliability of anomaly detection systems and raises critical ethical concerns. To balance fairness and utility in anomaly detection, several methods have been proposed (Deepak & Abraham, 2020; Song et al., 2021; Zhang & Davidson, 2021; Shekhar et al., 2021). These methods strive to optimize the balance between fairness and anomaly detection performance in the absence of ground truth labels, which presents a fundamental challenge in unsupervised anomaly detection. Nevertheless, they primarily focus on independent and identically distributed data, thereby overlooking the societal bias in graphs, which manifests in both node attributes and graph topology.

The bias in graphs poses a significant challenge to achieving fairness in graph-related tasks (Dai & Wang, 2021; Zhu et al., 2023). First, sensitive attributes are inherently spread across other attributes (Deepak & Abraham, 2020; Oh et al., 2022), so directly removing them is insufficient to ensure fairness (Neo et al., 2024). For example, the geographic location might correlate with religion and ethnicity. Second, since nodes with similar attributes are more likely to form connections, graph topology is also influenced by sensitive attributes (Rahman et al., 2019; Spinelli et al., 2021). Third, biased topology coupled with the message-passing mechanism in GNNs may inherit or even amplify the inherent bias in graphs (Dai & Wang, 2021; Wang et al., 2022; Zhu et al., 2023). Specifically, representations aggregated from neighboring nodes that share identical sensitive attributes may amplify features of the demographic group, thereby potentially affecting the fairness of the decision-making in GNNs.

Many efforts have been made to explore fairness for GNN-based methods (Li et al., 2021; Ma et al., 2022; Kim et al., 2022; Song et al., 2022). A common strategy involves eliminating sensitive information from the training graph and implementing the debiased graph for target tasks (Spinelli et al., 2021; Rahman et al., 2019; Dong et al., 2022). However, it is non-trivial to concurrently mitigate bias and preserve the integrity of anomalies, considering the overlap between the features of anomalies unrelated to sensitive attributes and the features of demographic groups linked to sensitive attributes. For instance, an edge might indicate that two nodes share the same sensitive attributes while displaying an anomalous connection. Another prevalent strategy involves training fair GNNs to perform the target task independently of sensitive attributes (Dai & Wang, 2021; Zhu et al., 2023). These methods are dedicated to end-to-end supervised node classification, where ground-truth labels are available. Thus, applying them to unsupervised GAD is challenging due to the absence of labels for anomaly detection.

In this paper, we propose a novel **D**is**E**ntangle-based **F**airn**E**ss-aware a**N**omaly **D**etection framework on attributed graphs, named **DEFEND**. In the first stage, to address the societal bias embedded in both node attributes and graph topology, we introduce disentangled fair representation learning on graphs to capture node representations that are both informative and independent of sensitive attributes. Specifically, the disentangled graph encoder can effectively separate sensitive-relevant and sensitive-irrelevant representations into independent subspaces with the guidance of a learnable adversary. In the second stage, given the absence of ground truth labels and the inherent complex bias in graph topology, we implement an additional decoder that reconstructs node attributes from the well-trained disentangled encoder, utilizing the reconstruction error as the anomaly score. To further alleviate discriminatory bias in detecting anomalies, we introduce a fairness constraint that enforces invariance of reconstruction errors across different sensitive attribute groups, effectively mitigating the influence of correlations between sensitive and the rest attributes. Our main contributions are summarized as follows:

- To the best of our knowledge, we proposed the first method DEFEND for fair unsupervised graph anomaly detection, which reduces discriminatory bias in anomaly detection.

- DEFEND employs constrained reconstruction error coupled with a disentangled graph encoder for fair anomaly detection on graphs.

- Extensive experiments on real-world datasets show that DEFEND achieves a competitive performance and significantly enhances fairness compared with baselines.

## 2 Preliminaries

### 2.1 Problem Definition

Let $\mathcal{G} = (\mathcal{V}, \mathbf{A}, \mathbf{X}, \mathbf{S})$ be an attributed graph with $N$ nodes and $E$ edges, where $\mathcal{V} = \{v_1, \ldots, v_N\}$ is the set of nodes. $\mathbf{y} \in \{0,1\}^N$ denotes the anomaly labels, where 1 indicates an anomalous node, and $\hat{\mathbf{y}}$ denotes the predicted labels. The adjacency matrix is denoted as $\mathbf{A} \in \{0,1\}^{N \times N}$, where $\mathbf{A}_{ij} = 1$ if there exists an edge between $v_i$ and $v_j$, otherwise, $\mathbf{A}_{ij} = 0$. $\mathbf{X} \in \mathbb{R}^{N \times d}$ represents the observed node attribute matrix, while $\mathbf{S} \in \mathbb{R}^{N \times m}$ represents the sensitive attribute matrix (e.g., gender, religion, ethnicity, etc.). $v_i$ and $v_j$ belong to the same demographic group if $s_i = s_j$. Here, $m$ is the total number of sensitive attributes. As described in (Deepak & Abraham, 2020; Sarhan et al., 2020; Oh et al., 2022), the sensitive attribute $\mathbf{S}$ is correlated with both the observed attribute $\mathbf{X}$ and labels $\mathbf{y}$ on many real-world datasets. The goal of fair graph anomaly detection is to provide unbiased prediction against sensitive attributes while achieving satisfactory accuracy simultaneously. To simplify the problem, in this work, we mainly focus on a single binary sensitive attribute, i.e., $\mathbf{S} \in \{0,1\}^{N \times 1}$. We can easily extend our method to more complicated settings as previous studies (Creager et al., 2019; Deepak & Abraham, 2020). More details are discussed in Section 3.3.

### 2.2 Fairness Metrics

Following (Agarwal et al., 2021; Wang et al., 2022; Zhu et al., 2023), we utilize two widely used metrics to evaluate the fairness of models among demographic groups.

**Demographic Parity** Dwork et al. (2012) dictates the equal predicted probability across demographic groups. It ensures predictions are statistically unbiased to sensitive attributes, e.g., if gender is a sensitive attribute, $\Delta_{DP}$ implies that the probability of individuals from different genders being classified as anomalous should be identical.

$$\Delta_{DP} = |P(\hat{y} = 1 | s = 0) - P(\hat{y} = 1 | s = 1)|, \tag{1}$$

where $\hat{y} \in \{0, 1\}$ is the predicted node label, and $\hat{y}_i = 1$ indicates node $v_i$ is a predicted anomaly.

**Equal Opportunity** (Hardt et al., 2016) requires the same true positive rates of identifying anomalies for each demographic group. Considering gender as a sensitive attribute, $\Delta_{EO}$ encourages individuals from different genders to have an equal probability of being correctly identified as anomalous. Given the ground-truth label $y \in \{0, 1\}$ where $y_i = 1$ denotes $v_i$ is a true anomaly, $\Delta_{EO}$ can be defined as:

$$\Delta_{EO} = |P(\hat{y} = 1 | s = 0, y = 1) - P(\hat{y} = 1 | s = 1, y = 1)|. \tag{2}$$

## 3 Proposed Method

We now introduce DEFEND, which aims to identify anomalies without skew towards demographic groups defined on sensitive attributes. The disentangled representation learning separates sensitive-relevant and sensitive-irrelevant representations in the latent space (see Section 3.1). Fair anomaly detection is achieved through constrained reconstruction error using the disentangled representations (see Section 3.2). We then discuss its generalization capability across diverse sensitive attributes in Section 3.3. We introduce the training and inference processes in Appendix A and provide a computational complexity analysis in Appendix B.

Figure 1 illustrates the overall workflow of DEFEND, which contains two major phases. Firstly, as shown in the left part of Figure 1, the disentangled fair representation learning phase separates sensitive-relevant representations $\mathbf{Z}_x$ and sensitive-irrelevant representations $\mathbf{Z}_s$ in the latent space. Specifically, a disentangled graph encoder $f_e$ maps the node attribute $\mathbf{X}$ and graph topology $\mathbf{A}$ into independent sensitive-relevant and -irrelevant subspaces. Subsequently, $f_a$ and $f_x$ decode adjacency matrix $\hat{\mathbf{A}}$ and node attributes $\hat{\mathbf{X}}$, respectively. Ideally, $\mathbf{Z}_x$ contains no sensitive information with a well-trained $f_e$. To promote the independence of $\mathbf{Z}_x$ from $\mathbf{Z}_s$, we include a learnable adversary $g_\omega$. Besides minimizing reconstruction error to obtain informative representations, we also endeavor to accurately infer sensitive attributes $\mathbf{S}$ from $\mathbf{Z}_s$. As shown in the right part of Figure 1, the unsupervised graph anomaly detection phase identifies anomalies based on sensitive-irrelevant representations $\mathbf{Z}_x$. However, the reconstruction of attributes and structures may lead

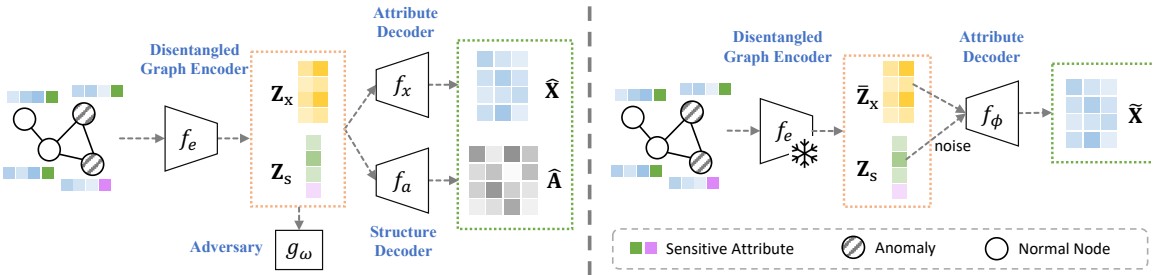

Figure 1: An overview of proposed DEFEND framework. (Left) Disentangled fair representation learning. The disentangled graph encoder $f_e$ can separate sensitive-irrelevant representations $\mathbf{Z}_x$ and sensitive-relevant representations $\mathbf{Z}_s$ in latent space. (Right) Reconstruct-based graph anomaly detection. The constrained reconstruction error between $\mathbf{X}$ and $\tilde{\mathbf{X}}$ are used to identify anomalies. ❄ means fixing model parameters.

to biased outcomes in decision-making, considering biases inherent in both attributes and structures. As such, the decoder $f_\phi$ only reconstructs node attributes from $\mathbf{Z}_x$ based on Multi-Layer Perceptron (MLP) without involving graph topology. Moreover, as node attributes $\mathbf{X}$ inherently have potential correlations with sensitive attributes $\mathbf{S}$, we further constrain the correlation between the reconstruction error and the predicted sensitive attributes.

## 3.1 Disentangled Fair Representation Learning

Informative sensitive-irrelevant representations are crucial for fair and accurate decision-making in GAD, which thoroughly considers features of nodes at both node and structural levels. However, since the potential bias in node attributes and graph topology, GNNs might amplify sensitive information when generating node representations with the message-passing mechanism (Dai & Wang, 2021; Zhu et al., 2023). Previous theoretical insights and empirical evidence have highlighted the effectiveness of disentangled representation learning in separating sensitive-irrelevant representations for augmenting fairness in downstream tasks like image classification (Creager et al., 2019; Oh et al., 2022). We posit that disentangled representation learning is also feasible to provide informative yet sensitive-irrelevant node representations for GAD. The details of disentangled fair representation learning on graph-structured data in DEFEND are described as follows.

### 3.1.1 Disentangled Graph Encoder

Based on the assumption that the latent space can be decomposed into two independent subspaces (Creager et al., 2019; Oh et al., 2022): one associated with sensitive attributes and the other devoid of them, the disentangled graph encoder $f_e$ strives to capture informative node representations that are irrelevant to sensitive attributes. For graph data, the posterior distribution of node representations $q(\mathbf{Z}_x, \mathbf{Z}_s|\mathbf{X}, \mathbf{A})$ is derived from node attributes $\mathbf{X}$ and graph topology $\mathbf{A}$. To achieve disentanglement, it is necessary to establish conditional independence between sensitive-irrelevant representations $\mathbf{Z}_x$ and sensitive-relevant representations $\mathbf{Z}_s$, given $\mathbf{X}$ and $\mathbf{A}$. The variational posterior distribution $q(\mathbf{Z}_x, \mathbf{Z}_s|\mathbf{X}, \mathbf{A})$ can be defined as the product of the individual distributions for $\mathbf{Z}_x$ and $\mathbf{Z}_s$:

$$q(\mathbf{Z}_x, \mathbf{Z}_s|\mathbf{X}, \mathbf{A}) = q(\mathbf{Z}_x|\mathbf{X}, \mathbf{A})q(\mathbf{Z}_s|\mathbf{X}, \mathbf{A}). \tag{3}$$

As the disentanglement is conducted in latent space, it is crucial to capture informative node representations from both node attributes and topological structure. Thus, we adopt GNNs as the backbone of the disentangled graph encoder $f_e$. In this work, we take Graph Convolutional Network (GCN) (Kipf & Welling, 2016) as an example. For the $l$-th convolutional layer, the node representation $\mathbf{H}^{(l)}$ is updated by:

$$\mathbf{H}^{(l)} = \text{Conv}(\mathbf{H}^{(l-1)}, \mathbf{A}) = \phi\left(\tilde{\mathbf{D}}^{-\frac{1}{2}}\tilde{\mathbf{A}}\tilde{\mathbf{D}}^{-\frac{1}{2}}\mathbf{H}^{(l-1)}\mathbf{W}^l\right), \tag{4}$$

where $\tilde{\mathbf{A}} = \mathbf{A} + \mathbf{I}$, $\tilde{\mathbf{D}}_{ii} = \sum_j \tilde{\mathbf{A}}_{ij}$, $\mathbf{I}$ is the identity matrix of $\mathbf{A}$, $\mathbf{W}^l$ is the weight matrix at the $l$-th layer and the initial node representation $\mathbf{H}^{(0)}$ is set to $\mathbf{X}$. Next, we use the reparameterization trick to estimate

the sensitive-irrelevant representations $\mathbf{Z}_x$ with GCN:

$$\mathbf{Z}_x = \boldsymbol{\mu} + \boldsymbol{\sigma} \cdot \boldsymbol{\varepsilon}, \ \boldsymbol{\varepsilon} \sim \mathcal{N}(\mathbf{0}, \mathbf{I}), \tag{5}$$

where $\mathcal{N}(\mathbf{0}, \mathbf{I})$ is the standard Gaussian distribution with a mean vector of zeros $\mathbf{0}$ and an identity covariance matrix $\mathbf{I}$. The mean matrix $\boldsymbol{\mu} = \mathrm{Conv}_{\boldsymbol{\mu}}(\mathrm{Conv}_{\mathrm{shr}}(\mathbf{X}, \mathbf{A}), \mathbf{A})$ and the log standard deviation matrix $\log \boldsymbol{\sigma} = \mathrm{Conv}_{\boldsymbol{\sigma}}(\mathrm{Conv}_{\mathrm{shr}}(\mathbf{X}, \mathbf{A}), \mathbf{A})$ are obtained by GCNs with a shared convolutional layer $\mathrm{Conv}_{\mathrm{shr}}$. Following (Creager et al., 2019), we estimate the sensitive-relevant representations $\mathbf{Z}_s$ in a deterministic manner:

$$\mathbf{Z}_s = \mathrm{Conv}_{\boldsymbol{\varphi}}(\mathrm{Conv}_{\mathrm{shr}}(\mathbf{X}, \mathbf{A}), \mathbf{A}). \tag{6}$$

where $\mathrm{Conv}_{\boldsymbol{\varphi}}$ is an additional layer for sensitive representations.

### 3.1.2 Decoders

In the decoding phase of disentangled representation learning, we reconstruct the attribute matrix $\mathbf{X}$, the adjacency matrix $\mathbf{A}$, and sensitive attributes $\mathbf{S}$. Due to sensitive-irrelevant representations $\mathbf{Z}_x$ are conditionally independent of the given sensitive attributes $\mathbf{S}$, the probability $p(\mathbf{S}|\mathbf{Z}_x, \mathbf{Z}_s)$ can be simplified to $p(\mathbf{S}|\mathbf{Z}_s)$. Consequently, the decoding phase can be formulated by:

$$p(\mathbf{X}, \mathbf{A}, \mathbf{S}|\mathbf{Z}_x, \mathbf{Z}_s) = p(\mathbf{X}|\mathbf{Z}_x, \mathbf{Z}_s)p(\mathbf{A}|\mathbf{Z}_x, \mathbf{Z}_s)p(\mathbf{S}|\mathbf{Z}_s). \tag{7}$$

We adopt an attribute decoder $f_x$ based on GNNs to model the distribution of reconstructed node attributes $p(\mathbf{X}|\mathbf{Z}_x, \mathbf{Z}_s)$. Besides, we consider an inner product decoder $f_a$ for structure reconstruction:

$$p(\mathbf{A}|\mathbf{Z}_x, \mathbf{Z}_s) = \mathrm{Sigmoid}(\mathbf{Z}\mathbf{Z}^\top), \tag{8}$$

where $\mathbf{Z} = \mathrm{Concat}(\mathbf{Z}_x, \mathbf{Z}_s)$. Next, to model the conditional probability distribution $p(\mathbf{S}|\mathbf{Z}_s)$, we employ a binary classifier, where sensitive attributes $\mathbf{S}$ are assumed to follow a Bernoulli distribution parameterized by the Sigmoid function applied to $\mathbf{Z}_s$. The conditional distribution for sensitive attributes is formulated as:

$$p(\mathbf{S}|\mathbf{Z}_s) = \mathrm{Bernoulli}(\mathbf{S}|\mathrm{Sigmoid}(\mathbf{Z}_s)). \tag{9}$$

### 3.1.3 Adversary

To encourage the independence between sensitive-irrelevant representations $\mathbf{Z}_x$ and sensitive-relevant representations $\mathbf{Z}_s$, it is imperative that the aggregate posterior distribution can be factorized as $q(\mathbf{Z}_x, \mathbf{Z}_s) = q(\mathbf{Z}_x)q(\mathbf{Z}_s)$. Thus, we employ the Kullback-Leibler (KL) divergence between $q(\mathbf{Z}_x, \mathbf{Z}_s)$ and $q(\mathbf{Z}_x)q(\mathbf{Z}_s)$ as the disentanglement criteria, which can be expressed as $KL\big[q(\mathbf{Z}_x, \mathbf{Z}_s) \,\|\, q(\mathbf{Z}_x)q(\mathbf{Z}_s)\big]$. Following (Kim & Mnih, 2018; Sugiyama et al., 2012; Creager et al., 2019), we adopt a binary adversary $g_\omega$ to encourage the disentanglement by approximating the log density ratio inherent in the KL divergence term:

$$
\begin{aligned}
KL\big[q(\mathbf{Z}_x, \mathbf{Z}_s) \,\|\, q(\mathbf{Z}_x)q(\mathbf{Z}_s)\big] &= \mathbb{E}_{q(\mathbf{Z}_x, \mathbf{Z}_s)} \log \frac{q(\mathbf{Z}_x, \mathbf{Z}_s)}{q(\mathbf{Z}_x)q(\mathbf{Z}_s)} \\
&\approx \mathbb{E}_{q(\mathbf{Z}_x, \mathbf{Z}_s)}[\log p(\tilde{\mathbf{y}} = 1|\mathbf{Z}_x, \mathbf{Z}_s) - \log p(\tilde{\mathbf{y}} = 0|\mathbf{Z}_x, \mathbf{Z}_s)],
\end{aligned}
\tag{10}
$$

where $\tilde{\mathbf{y}} = 1$ denotes "true" samples from the aggregate posterior $q(\mathbf{Z}_x, \mathbf{Z}_s)$ and $\tilde{\mathbf{y}} = 0$ denotes "fake" samples from the product of the marginals $q(\mathbf{Z}_x)q(\mathbf{Z}_s)$. Specifically, we implement an MLP as $g_\omega$ to predict whether the sensitive-irrelevant representation $\mathbf{z}_x^i$ and the sensitive-relevant representation $\mathbf{z}_s^j$ are from the same node.

### 3.1.4 Learning Objective

We optimize the disentangled graph encoder $f_e$, structure decoder $f_a$, and attribute decoder $f_x$ from three aspects, including the variational lower bound term, disentanglement term, and predictiveness term. First, the variational lower bound term comprises a reconstruction term and a KL divergence regularization term. The reconstruction term is defined as:

$$\mathcal{L}_{rec} = \mathbb{E}_{q(\mathbf{Z}_x, \mathbf{Z}_s|\mathbf{X}, \mathbf{A})}\left[(1 - \epsilon)\log p(\mathbf{X}|\mathbf{Z}_x, \mathbf{Z}_s) + \epsilon \log p(\mathbf{A}|\mathbf{Z}_x, \mathbf{Z}_s)\right], \tag{11}$$

where the first term is the reconstruction error for node attributes and the second term is for graph topology. Moreover, $\epsilon = \frac{\sigma_{\mathbf{X}}}{\sigma_{\mathbf{X}}+\sigma_{\mathbf{A}}}$ is the weight coefficient for automated balancing the impact of structure and attribute reconstruction (Liu et al., 2022b), where $\sigma_{\mathbf{A}}$ and $\sigma_{\mathbf{X}}$ denotes the standard deviations of $\mathbf{A}$ and $\mathbf{X}$, respectively. The KL divergence regularization term (Kingma, 2013) is employed to minimize the KL divergence between the posterior distribution $q(\mathbf{Z}_x, \mathbf{Z}_s | \mathbf{X}, \mathbf{A})$ and the prior distribution $p(\mathbf{Z}_x, \mathbf{Z}_s)$. Thus, the variational lower bound term can be formulated as:

$$\mathcal{L}_{vae} = \mathcal{L}_{rec} - KL[q(\mathbf{Z}_x, \mathbf{Z}_s | \mathbf{X}, \mathbf{A}) \| p(\mathbf{Z}_x, \mathbf{Z}_s)], \tag{12}$$

where $p(\mathbf{Z}_x, \mathbf{Z}_s) = p(\mathbf{Z}_x)p(\mathbf{Z}_s)$ under the assumption that $\mathbf{Z}_x$ and $\mathbf{Z}_s$ are independent. The prior distributions $p(\mathbf{Z}_x)$ and $p(\mathbf{Z}_s)$ are modeled by the standard Gaussian distribution and uniform distribution, respectively. Next, the disentanglement term, which encourages the separation of sensitive-relevant and -irrelevant representations as detailed in Eq. (10), is calculated as follows:

$$\mathcal{L}_{dis} = \mathbb{E}_{\mathbf{z}_x^i, \mathbf{z}_s^i \sim q(\mathbf{Z}_x, \mathbf{Z}_s)} \log p(\tilde{y} = 1 | \mathbf{z}_x^i, \mathbf{z}_s^i) - \log p(\tilde{y} = 0 | \mathbf{z}_x^i, \mathbf{z}_s^i). \tag{13}$$

Intuitively, accurate prediction of sensitive attributes can enhance the comprehensive understanding of node attributes in latent space and facilitate a clearer distinction between representations associated with sensitive attributes and those that are not. Thus, DEFEND incorporates a predictiveness term to align the sensitive representation $\mathbf{Z}_s$ closely with the given sensitive attributes $\mathbf{S}$:

$$\mathcal{L}_{pre} = \mathbb{E}_{q(\mathbf{Z}_s | \mathbf{X}, \mathbf{A})} \log p(\mathbf{S} | \mathbf{Z}_s). \tag{14}$$

The overall loss for optimizing $f_e$, $f_a$ and $f_x$ can be defined as:

$$\mathcal{L}_{total} = \mathcal{L}_{vae} + \gamma \mathcal{L}_{dis} + \alpha \mathcal{L}_{pre}, \tag{15}$$

where $\alpha$ and $\gamma$ are the weight coefficients to control the impact of the predictiveness and disentanglement terms relative to the variational lower bound term, respectively.

To train the binary adversary $g_\omega$, the true sample $(\mathbf{z}_x^i, \mathbf{z}_s^i)$ is sampled from the aggregate posterior $q(\mathbf{Z}_x, \mathbf{Z}_s)$ while the fake sample $(\mathbf{z}_x^j, \mathbf{z}_s^k)$ is sampled from the product of marginal posterior distributions $q(\mathbf{Z}_x)q(\mathbf{Z}_s)$. The adversarial loss is formulated as:

$$\mathcal{L}_{adv} = \mathbb{E}_{\mathbf{z}_x^i, \mathbf{z}_s^i \sim q(\mathbf{Z}_x, \mathbf{Z}_s)} \log p(\tilde{y} = 1 | \mathbf{z}_x^i, \mathbf{z}_s^i) + \mathbb{E}_{\mathbf{z}_x^j, \mathbf{z}_s^k \sim q(\mathbf{Z}_x)q(\mathbf{Z}_s)} \log[1 - p(\tilde{y} = 0 | \mathbf{z}_x^j, \mathbf{z}_s^k)]. \tag{16}$$

where $\tilde{y} = 1$ denotes true samples and $\tilde{y} = 0$ denotes fake samples.

The optimizations of $f_e$, $f_a$ and $f_x$ using $\mathcal{L}_{total}$ and $g_\omega$ using $\mathcal{L}_{adv}$ are conducted adversarially. A well-trained disentangled graph encoder $f_e$, which effectively separates sensitive-irrelevant representations from node attributes and graph topology, can be adeptly employed in downstream tasks for enhancing fairness.

## 3.2 Graph Anomaly Detection

Next, the primary objective is to detect anomalies unbiased to any demographic groups based on deterministic sensitive-irrelevant representations $\bar{\mathbf{Z}}_x = \boldsymbol{\mu}$. Since anomalies typically significantly deviate from the majority of nodes, reconstruction error has been widely used to measure anomaly scores (Ding et al., 2019; Fan et al., 2020; Shekhar et al., 2021). Considering the potential bias in graph topology as demonstrated in previous studies (Rahman et al., 2019; Spinelli et al., 2021; Zhu et al., 2023), we solely reconstruct node attributes and employ MLP as the backbone of the attribute decoder $f_\phi$ to mitigate the impact of biased topology during the message-passing process. The anomaly score $o_i$ is evaluated by reconstructing node attributes $\mathbf{X}$ from sensitive-irrelevant representations $\bar{\mathbf{Z}}_x$:

$$o_i = ||\mathbf{x}_i - \tilde{\mathbf{x}}_i||_F^2, \tag{17}$$

where $\tilde{\mathbf{X}} = f_\phi(\bar{\mathbf{Z}}_x, \tilde{\mathbf{Z}}_s)$ and $\tilde{\mathbf{Z}}_s$ denotes a shuffled variant of $\mathbf{Z}_s$. Additionally, since sensitive attributes are correlated with observed node attributes (Deepak & Abraham, 2020; Sarhan et al., 2020; Oh et al., 2022) and $\bar{\mathbf{Z}}_x$ is supposed to be devoid of sensitive information, the reconstruction error inevitably correlates with

sensitive attributes. To mitigate the impact of this correlation and prevent directly leveraging sensitive attributes, we propose a correlation constraint term, which measures the absolute correlation between the reconstruction error $o_i$ and predicted sensitive attributes $\mathbf{z}_s^i$:

$$\mathcal{L}_{corr} = \left| \frac{(\sum_{i \in \mathcal{V}} o_i - \mu_o)(\sum_{i \in \mathcal{V}} \mathbf{z}_s^i - \mu_{zs}))}{\sigma_o \sigma_{zs}} \right|, \tag{18}$$

where $\mu_o$ and $\mu_{zs}$ are the corresponding means, while $\sigma_o$ and $\sigma_{zs}$ are the standard deviations of $\mathbf{o}$ and $\mathbf{Z}_s$.

### 3.2.1 Learning Objective

In the anomaly detection phase, the encoder $f_e$ is set to be non-trainable, as it has already mastered the separation of sensitive-relevant and -irrelevant representations during the disentangled representation learning phase. Accordingly, the optimization is exclusively concentrated on the decoder $f_\phi$. The overall loss consisting of the reconstruction term and the correlation constraints term can be formulated as:

$$\mathcal{L}_{ad} = \mathcal{L}_{rec}^X + \beta \mathcal{L}_{corr}, \tag{19}$$

where $\mathcal{L}_{rec}^X = \sum_i^n o_i$ is the attribute reconstruction loss and $\beta$ is the penalty of sensitive information in $\mathbf{X}$.

### 3.3 Discussion

We analyze the generalization of DEFEND on various types of sensitive attributes. **(1) Multiple Binary Sensitive Attributes.** Let $\mathbf{s} = \left[ s^0, s^1, \ldots, s^{m-1} \right]$ denotes the sensitive attributes for each node, where $s^j \in \{0,1\} (0 \leq j < m)$. The disentanglement between sensitive-irrelevant representations $\mathbf{Z}_x$ and sensitive-relevant representations $\mathbf{Z}_s \in \mathbb{R}^{N \times m}$ requires that $\mathbf{Z}_x$ is independent of each sensitive attribute $\mathbf{Z}_s^{(j)}$. Thus, $\mathbf{Z}_x$ and $\mathbf{Z}_s$ are disentangled if the aggregate posterior distribution can be factorized as $p(\mathbf{Z}_x, \mathbf{Z}_s) = p(\mathbf{Z}_x) \prod_j p(\mathbf{Z}_s^{(j)})$. **(2) Single Categorical/Continuous Sensitive Attributes.** With single binary sensitive attributes, the conditional distribution $p(\mathbf{S}|\mathbf{Z}_s)$ is typically modeled by a Bernoulli distribution. If with categorical attributes, it shifts to a Multinomial distribution, i.e. $p(\mathbf{S}|\mathbf{Z}_s) = \text{Multinomial}(\mathbf{S}|\text{SoftMax}(\mathbf{Z}_s))$. Similarly, for continuous attributes, $p(\mathbf{S}|\mathbf{Z}_s)$ can be modeled by a Gaussian distribution.

## 4 Experiments

### 4.1 Experimental Setup

**Datasets.** We employ three real-world datasets for fair GAD, which provide both real sensitive attributes and ground-truth labels for GAD. In Reddit and Twitter (Neo et al., 2024) datasets, the sensitive attribute is the political leaning of users, while the anomaly label is assigned to misinformation spreaders. The Credit (Agarwal et al., 2021) dataset focuses on payment default detection, with age as the sensitive attribute. Details of these datasets are summarized in Table 1. More details are introduced in Appendix C.

Table 1: Statistics of datasets. $\rho_G$ denotes the ratio of the minority and majority group and $\rho_A$ denotes the ratio of the anomalies and normal nodes.

| Dataset | # Nodes | # Edges | # Attributes | $\rho_G$ | $\rho_A$ | Sensitive Attributes | Anomaly Labels |
|---|---|---|---|---|---|---|---|
| **Reddit** | 9,892 | 1,211,748 | 385 | 0.1502 | 0.1584 | Political leaning | Misinformation spreader |
| **Twitter** | 47,712 | 468,697 | 780 | 0.1365 | 0.0713 | Political leaning | Misinformation spreader |
| **Credit** | 30,000 | 1,436,858 | 13 | 0.0983 | 0.2840 | Age | Payment default |

**Baselines.** We compare DEFEND with (1) GAD methods, including **DOMINANT** (Ding et al., 2019), **CoLA** (Liu et al., 2021), **CONAD** (Xu et al., 2022), and **VGOD** (Huang et al., 2023); (2) GAD methods augmented with Fairness Regularizers like **FairOD** (Shekhar et al., 2021), **HIN** (Zeng et al., 2021), and **Correlation** (Shekhar et al., 2021), which incorporate fairness constraints into optimization process; (3) GAD methods operated on graphs pre-processed by Graph Debiasers, such as **FairWalk** (Rahman et al., 2019) and **EDITS** (Dong et al., 2022). The details of baselines are introduced in Appendix D.

Table 2: Comparison results of DEFEND against all baseline methods on Twitter. ↑ denotes larger values are better, whereas ↓ denotes lower values are preferable. The best and second best performances are highlighted in **bold** and underlined, respectively.

| | | AUC-ROC ↑ | AUC-PR ↑ | $\Delta_{DP}$ ↓ | $\Delta_{EO}$ ↓ | Comp. | Avg. Rank |
|---|---|---|---|---|---|---|---|
| - | DOMINANT | 56.49±0.61 | 8.92±0.15 | 4.33±0.47 | 4.51±0.44 | -29.36 | 13.25 |
| | CoLA | 43.74±1.08 | 5.23±0.15 | 2.63±0.32 | 2.47±1.08 | -42.06 | 18.375 |
| | CONAD | 56.12±0.73 | 8.83±0.17 | 4.08±0.47 | 4.48±0.35 | -29.54 | 13.25 |
| | VGOD | 73.59±0.31 | 16.02±0.72 | 12.56±1.01 | 11.49±1.84 | -20.37 | 14 |
| FairWalk | DOMINANT | 53.06±0.81 | 8.17±0.43 | 1.32±0.27 | 1.27±0.35 | -27.29 | 11.75 |
| | CoLA | 49.02±0.54 | 6.34±0.23 | 0.23±0.14 | 0.34±0.16 | -31.14 | 10.75 |
| | CONAD | 53.36±0.70 | 8.32±0.42 | 1.30±0.18 | 1.37±0.39 | -26.92 | 11 |
| | VGOD | 60.21±0.26 | 9.14±0.15 | 9.84±0.27 | 4.98±0.37 | -31.40 | 14.25 |
| EDITS | DOMINANT | 53.79±0.25 | 8.75±0.06 | 2.63±0.05 | 2.21±0.15 | -28.23 | 12.75 |
| | CoLA | 46.62±1.35 | 5.70±0.30 | 1.84±0.39 | 1.03±0.82 | -36.48 | 14.25 |
| | CONAD | 53.83±0.24 | 8.75±0.06 | 2.62±0.03 | 2.16±0.13 | -28.13 | 11.875 |
| | VGOD | 82.06±0.87 | 25.75±1.40 | 20.43±1.42 | 22.32±2.45 | -20.87 | 13 |
| FairOD | DOMINANT | 53.69±3.63 | 7.87±1.10 | 2.56±1.51 | 2.49±1.52 | -29.42 | 14.25 |
| | CoLA | 45.81±7.40 | 6.29±1.52 | 1.95±1.82 | 1.23±0.77 | -37.01 | 14.625 |
| | CONAD | 57.09±0.43 | 9.08±0.13 | 4.48±0.45 | 4.59±0.56 | -28.83 | 13.75 |
| | VGOD | 76.27±1.12 | 15.56±0.54 | 9.75±1.07 | 5.61±1.67 | -9.46 | 12.5 |
| HIN | DOMINANT | 54.07±2.22 | 8.23±0.79 | 2.50±1.24 | 3.51±0.96 | -29.64 | 13 |
| | CoLA | 48.08±5.31 | 6.49±1.18 | 2.38±1.23 | 0.62±0.69 | -34.36 | 12.75 |
| | CONAD | 53.74±2.74 | 8.12±1.02 | 2.59±1.36 | 3.34±1.04 | -30.00 | 14.25 |
| | VGOD | 81.58±1.88 | 18.86±1.00 | 12.52±0.91 | 9.32±1.38 | -7.33 | 12.5 |
| Correlation | DOMINANT | 56.21±0.60 | 8.81±0.14 | 4.22±0.38 | 4.58±0.29 | -29.71 | 14.25 |
| | CoLA | 48.65±2.89 | 6.29±0.48 | 3.32±1.18 | 3.42±2.10 | -37.73 | 17.875 |
| | CONAD | 55.94±0.71 | 8.76±0.17 | 4.00±0.44 | 4.52±0.31 | -29.75 | 14.25 |
| | VGOD | 71.78±0.58 | 11.80±0.24 | 5.22±0.56 | 0.76±0.53 | -8.33 | 8.75 |
| Ours | DEFEND | 75.59±2.00 | 11.85±0.79 | 0.72±0.70 | 0.79±0.61 | **0** | **3.75** |

**Evaluation metrics.** Following (Ding et al., 2019; Liu et al., 2021; Xu et al., 2022; Chai et al., 2022; Neo et al., 2024), we evaluate the anomaly detection performance with the Area Under the Receiver Operating Characteristic Curve (**AUC-ROC**) and the Area Under the Precision-Recall Curve (**AUC-PR**). Higher AUC-ROC and AUC-PR indicate superior anomaly detection capabilities. Regarding demographic fairness, we adopt Demographic Parity ($\mathbf{\Delta_{DP}}$) and Equal Opportunity ($\mathbf{\Delta_{EO}}$) following (Dai & Wang, 2021; Dong et al., 2022). The concepts are detailed in Section 2.2. Lower $\Delta_{DP}$ and $\Delta_{EO}$ suggest better fairness. To quantitatively exhibit the performance trade-offs, we introduce **Average Rank** of two accuracy metrics and two fairness metrics (Wang et al., 2022; Guo et al., 2023) and **Composite Score** that reflects the combined performance improvement relative to our proposed DEFEND across all metrics.

**Implementation details.** For DEFEND, we use 2-layer GCN for the disentangled graph encoder $f_e$ and attribute decoder $f_x$, and 2-layer MLP for the decoder $f_\phi$ in the anomaly detection phase. The hidden dimension of each layer is fixed to be 64. We employ the Adam optimizer with a learning rate set to 0.001 for Reddit and Credit datasets and 0.005 for Twitter. We set the maximum training epoch in disentangled representation learning as 100, and adopt an early stopping strategy when the loss does not decrease for 20 epochs. In the anomaly detection phase, we train the decoder $f_\phi$ for 100 epochs. We tune $\alpha$ and $\gamma$ in Equation 15 from $\{0.1, 0.5, 1.0, 1.5, 2.0, 2.5\}$, and the weight of correlation constrains $\beta$ in Equation 19 from $\{1e-15, 5e-15, 1e-10, 5e-10, 1e-9\}$, respectively. We conduct all experiments on one Linux server with an NVIDIA TESLA A800 GPU (80 GB RAM). We run all methods ten times and report the average results to prevent extreme cases. More implementation details of baselines are described in Appendix E.

## 4.2 Performance Comparison

### 4.2.1 Fairness and Utility Performance

Table 2, Table 3 and Table 4 show the fairness ($\Delta_{DP}$ and $\Delta_{EO}$) and utility (AUC-ROC and AUC-PR) performance of DEFEND and all baselines on Twitter and Reddit, respectively. We have the following ob-

Table 3: Comparison results of DEFEND against all baseline methods on Reddit. ↑ denotes larger values are better, whereas ↓ denotes lower values are preferable. The best and second best performances are highlighted in **bold** and underlined, respectively. OOM indicates out of memory.

| | | AUC-ROC ↑ | AUC-PR ↑ | $\Delta_{DP}$ ↓ | $\Delta_{EO}$ ↓ | Comp. | Avg. Rank |
|---|---|---|---|---|---|---|---|
| - | DOMINANT | 60.82±0.09 | 20.02±0.04 | 13.20±0.08 | 5.59±0.19 | -13.01 | 10.25 |
| | CoLA | 45.20±0.98 | 17.90±1.54 | 4.95±1.78 | 4.06±1.85 | -20.97 | 12.5 |
| | CONAD | 60.81±0.10 | 20.02±0.05 | 13.32±0.33 | 5.70±0.37 | -13.25 | 11.5 |
| | VGOD | 72.01±0.93 | 39.38±2.45 | 42.47±5.61 | 46.79±6.09 | -52.93 | 12 |
| FairWalk | DOMINANT | 51.26±1.08 | 14.55±0.65 | 2.45±0.79 | 1.97±1.25 | -13.67 | 11 |
| | CoLA | 51.12±0.83 | 14.83±0.85 | 0.69±0.37 | 0.60±0.59 | -10.40 | 10 |
| | CONAD | 51.26±1.88 | 14.68±1.15 | 2.49±1.28 | 2.52±1.58 | -14.13 | 11.25 |
| | VGOD | 67.08±0.61 | 28.53±0.28 | 31.99±0.53 | 29.89±1.02 | -41.33 | 11.75 |
| EDITS | DOMINANT | OOM | OOM | OOM | OOM | - | - |
| | CoLA | 54.41±1.62 | 23.13±3.67 | 23.74±5.76 | 21.00±5.38 | -42.26 | 14 |
| | CONAD | OOM | OOM | OOM | OOM | - | - |
| | VGOD | OOM | OOM | OOM | OOM | - | - |
| FairOD | DOMINANT | 60.94±0.07 | 19.97±0.08 | 13.08±0.21 | 5.04±0.15 | -12.27 | 9.5 |
| | CoLA | 45.76±10.23 | 15.77±5.42 | 13.26±6.24 | 5.98±5.83 | -32.77 | 16.5 |
| | CONAD | 60.53±0.11 | 19.53±0.08 | 12.57±0.11 | 5.02±0.11 | -12.59 | 11 |
| | VGOD | 71.65±2.55 | 30.40±4.98 | 28.66±7.70 | 37.74±4.28 | -39.41 | 11.5 |
| HIN | DOMINANT | 60.91±0.06 | 20.10±0.04 | 13.38±0.10 | 5.69±0.22 | -13.12 | 10.5 |
| | CoLA | 45.52±1.47 | 18.59±1.68 | 4.58±2.50 | 4.43±2.28 | -19.96 | 12.25 |
| | CONAD | 60.89±0.18 | 20.06±0.10 | 13.35±0.21 | 5.81±0.45 | -13.27 | 11.25 |
| | VGOD | 72.66±3.32 | 26.18±3.36 | 17.92±6.12 | 30.39±9.38 | -24.53 | 10.75 |
| Correlation | DOMINANT | 60.38±0.08 | 19.57±0.06 | 12.27±0.25 | 4.42±0.20 | -11.80 | 10.25 |
| | CoLA | 50.94±6.37 | 15.29±3.50 | 9.63±7.26 | 11.68±8.23 | -30.14 | 15.25 |
| | CONAD | 59.51±0.33 | 18.86±0.28 | 10.94±0.74 | 3.80±0.35 | -11.43 | 10.25 |
| | VGOD | 75.68±3.07 | 31.09±2.83 | 32.42±9.53 | 42.72±9.65 | -43.43 | 11.25 |
| Ours | DEFEND | 60.67±0.42 | 16.46±0.25 | 0.94±0.72 | 1.13±0.81 | **0** | **8** |

Figure 2: Fairness-Utility trade-off curves of different methods on three datasets. The upper-left corner is optimal, which has high AUC-ROC and low $\Delta_{EO}$.

servations: **(1) DEFEND demonstrates superior performance in balancing fairness and accuracy trade-offs.** To evaluate this trade-off quantitatively, we analyze the average ranks and relative improvements across two accuracy metrics and two fairness metrics. The runner-up performing model shows a 7.33% decline in performance compared to DEFEND, with DEFEND achieving an average rank of 3.75 versus 8.75 for the runner-up model on Twitter. Figure 2 illustrates the fairness-utility trade-off curve, which further validates the superiority of DEFEND (detailed analysis provided in Section 4.2.2). **(2) Standard GAD methods exhibit a clear trade-off between accuracy and fairness.** For instance, VGOD with Correlation achieves superior detection performance with AUC-ROC reaching 75.68% and AUC-PR attaining 31.09%, but demonstrates substantial fairness violations with $\Delta_{DP}$ at 32.42% and $\Delta_{EO}$ at 42.72% on Reddit. This observation reveals an inherent tension between detection capability and fairness in existing approaches.

Table 4: Comparison results of DEFEND against all baseline methods on Credit. ↑ denotes larger values are better, whereas ↓ denotes lower values are preferable. The best and second best performances are highlighted in **bold** and underlined, respectively.

| | | AUC-ROC ↑ | AUC-PR ↑ | $\Delta_{DP}$ ↓ | $\Delta_{EO}$ ↓ | Comp. | Avg. Rank |
|---|---|---|---|---|---|---|---|
| - | DOMINANT | 52.34±4.14 | 25.28±4.33 | 1.63±1.00 | 2.34±1.62 | -2.11 | 8.75 |
| | CoLA | 46.31±0.94 | 18.96±0.54 | 3.04±1.22 | 1.48±1.26 | -15.01 | 16 |
| | CONAD | 53.68±5.28 | 26.02±5.03 | 3.91±2.77 | 4.19±2.61 | -4.16 | 10.75 |
| | VGOD | 54.81±0.90 | 26.02±0.86 | 17.54±1.83 | 13.59±1.84 | -26.06 | 14.75 |
| FairWalk | DOMINANT | 49.56±0.40 | 21.78±0.44 | 0.67±0.74 | 1.50±0.99 | -6.60 | 11.75 |
| | CoLA | 50.29±0.17 | 22.63±0.31 | 0.32±0.28 | 0.22±0.23 | -6.32 | 8 |
| | CONAD | 49.76±0.55 | 21.91±0.63 | 1.21±0.83 | 1.02±0.80 | -3.38 | 10.25 |
| | VGOD | 49.33±0.10 | 22.00±0.06 | 31.57±0.94 | 29.43±1.17 | -65.44 | 22 |
| EDITS | DOMINANT | 49.19±1.71 | 22.06±1.85 | 2.99±2.12 | 2.73±1.59 | -10.23 | 14.5 |
| | CoLA | 49.82±1.01 | 21.81±0.73 | 5.07±2.79 | 2.95±2.40 | -13.54 | 16.25 |
| | CONAD | 49.60±1.50 | 22.36±1.14 | 5.23±4.18 | 4.51±2.66 | -12.14 | 17 |
| | VGOD | 46.61±3.23 | 20.52±1.61 | 9.32±10.38 | 9.27±9.20 | -27.23 | 21.75 |
| FairOD | DOMINANT | 52.59±4.01 | 24.64±4.29 | 4.37±3.54 | 5.04±3.67 | -7.95 | 13.5 |
| | CoLA | 47.43±6.33 | 21.24±4.40 | 7.99±5.79 | 6.09±4.93 | -21.18 | 20.25 |
| | CONAD | 55.03±5.39 | 27.38±5.14 | 4.10±3.59 | 4.02±3.66 | -1.48 | 8.5 |
| | VGOD | 55.70±1.32 | 26.87±1.39 | 16.76±1.29 | 13.02±1.35 | -22.98 | 13 |
| HIN | DOMINANT | 51.98±4.27 | 24.88±4.38 | 1.81±1.28 | 2.40±1.60 | -3.11 | 10 |
| | CoLA | 46.31±0.94 | 18.96±0.54 | 3.06±1.20 | 1.45±1.28 | -15.01 | 15.5 |
| | CONAD | 53.77±3.69 | 25.40±3.54 | 2.73±1.48 | 2.66±1.71 | -1.99 | 8.25 |
| | VGOD | 57.26±3.19 | 28.15±2.66 | 13.54±4.17 | 11.11±3.78 | -15.00 | 11.5 |
| Correlation | DOMINANT | 52.37±4.11 | 25.33±4.28 | 1.55±0.99 | 2.27±1.62 | -1.87 | 7.75 |
| | CoLA | 50.58±7.68 | 23.33±5.34 | 4.92±3.18 | 6.32±4.14 | -13.10 | 15.75 |
| | CONAD | 53.68±5.26 | 26.02±5.03 | 3.87±2.78 | 4.13±2.55 | -4.07 | 10.25 |
| | VGOD | 55.86±1.10 | 26.85±0.99 | 12.42±1.42 | 8.27±1.90 | -13.75 | 11.75 |
| Ours | DEFEND | 55.41±1.80 | 23.70±0.45 | 1.78±1.12 | 1.57±1.00 | **0.00** | **7.25** |

**(3) The integration of graph debiasers and fairness regularizers typically improves fairness.** For example, DOMINANT with FairWalk achieves better fairness with $\Delta_{DP}$ of 2.45% and $\Delta_{EO}$ of 1.97%, while DOMINANT with FairOD achieves better fairness with $\Delta_{DP}$ of 13.08% and $\Delta_{EO}$ of 5.04% on Reddit. However, these fairness improvements sometimes comes at the cost of reduced detection performance, e.g. AUC-ROC of DOMINANT with FairWalk dropping from 60.82% to 51.26% on Reddit.

### 4.2.2 Trade-off between Fairness and Utility

We extend our analysis to a comparative evaluation of the fairness-utility trade-off performance of DEFEND with GAD methods incorporating fairness regularizers. We choose $\Delta_{EO}$ to measure fairness and AUC-ROC for utility assessment. Each method is trained on a range of hyperparameters and the resulting Pareto front curves are presented in Figure 2. Notably, the optimal point is at the upper-left corner, which has perfect accuracy and fairness, reflected by a high AUC-ROC coupled with a low $\Delta_{EO}$. Specifically, a high AUC-ROC signified proficient detection of both normal and anomalous nodes, whereas a low $\Delta_{EO}$ indicates an equivalent probability of correctly identifying anomalies across different demographic groups. In contrast, the upper-right corner represents high performance at the cost of poor fairness, while the lower-left exhibits strong fairness but compromised performance. From Figure 2, we can observe the trade-off curve of DEFEND exhibits a superior distribution near the optimal point than the baselines.

### 4.3 Ablation Study

We conduct ablation studies to evaluate the effectiveness of key components of DEFEND. We verify the performance of four variants of DEFEND: (1) DEFEND without correlation constraints term in Equation 19 in anomaly detection (**DEFEND-C**). (2) DEFEND without the disentangled representation learning (**DEFEND-D**). (3) DEFEND without the adversary $g_\omega$ in disentangled representation learning (**DEFEND-**

Table 5: Comparison results of DEFEND and its variants. ↑ denotes larger values are better, whereas ↓ denotes lower values are preferable. The best and second best performances are highlighted in **bold** and underlined, respectively.

| Dataset | Method | AUC-ROC ↑ | AUC-PR ↑ | $\Delta_{DP}$ ↓ | $\Delta_{EO}$ ↓ | Comp. | Avg. Rank |
|---|---|---|---|---|---|---|---|
| Reddit | DEFEND-D | **64.41±1.23** | **20.50±0.82** | 14.36±1.23 | 13.57±1.66 | -18.08 | 3 |
| | DEFEND-C | 64.54±0.75 | 20.43±0.45 | 13.95±1.79 | 12.51±3.03 | -16.55 | 3 |
| | DEFEND-A | 60.44±1.06 | 16.39±0.41 | 1.64±1.36 | 2.53±1.99 | -2.4 | 3.5 |
| | DEFEND+S | 54.74±1.02 | 14.78±0.33 | **0.90±0.54** | 1.50±1.39 | -7.94 | 3.25 |
| | DEFEND | 60.67±0.42 | 16.46±0.25 | 0.94±0.72 | **1.13±0.81** | **0.00** | **2.25** |
| Twitter | DEFEND-D | 87.44±0.14 | 23.63±0.19 | 15.53±0.22 | 13.85±1.20 | -4.24 | 3 |
| | DEFEND-C | **87.58±0.16** | **23.85±0.27** | 15.86±0.47 | 15.31±0.97 | -5.67 | 3 |
| | DEFEND-A | 83.29±4.70 | 18.83±5.34 | 8.89±7.18 | 8.37±7.03 | -1.07 | 3 |
| | DEFEND+S | 50.15±0.74 | 6.48±0.13 | 1.28±0.19 | 1.59±1.21 | -32.17 | 3.5 |
| | DEFEND | 75.59±2.00 | 11.85±0.79 | **0.72±0.70** | **0.79±0.61** | **0.00** | **2.5** |
| Credit | DEFEND-D | **67.60±0.19** | **36.12±0.27** | 18.98±0.42 | 22.04±0.94 | -13.06 | 3 |
| | DEFEND-C | 67.23±1.04 | 35.83±1.13 | 18.42±1.07 | 20.96±1.77 | -12.08 | 3 |
| | DEFEND-A | 54.69±1.27 | 23.47±0.36 | 1.56±0.79 | 2.45±1.35 | -1.60 | 3.25 |
| | DEFEND+S | 49.08±1.35 | 21.65±0.75 | **1.33±1.61** | 1.66±1.25 | -8.03 | 3.25 |
| | DEFEND | 55.41±1.80 | 23.70±0.45 | 1.78±1.12 | **1.57±1.00** | **0.00** | **2.5** |

**A**). (4) DEFEND with a dot product decoder to reconstruct the graph structure for anomaly detection (**DEFEND+S**). The results are shown in Table 5.

The following observations can be made from Table 5. **(1)** DEFEND outperforms all variants in fairness, which validates the importance of each component in promoting fairness for GAD. Nevertheless, the accuracy of DEFEND is surpassed by several variants, such as DEFEND-C and DEFEND-D, likely due to the stringent fairness requirements. As illustrated in Figure 2 (b), relaxing the fairness constraints can improve utility performance. Specifically, the trade-off curve shows that DEFEND achieves an AUC-ROC of nearly 80% with a $\Delta_{EO}$ of about 1.5%. In this case, DEFEND not only exceeds its variants in fairness but also maintains a comparable utility performance to them. **(2)** The fairness of DEFEND-C is significantly worse than DEFEND, indicating a notable correlation between reconstruction error and sensitive attributes. **(3)** The fairness performance of DEFEND-D further drops than DEFEND, highlighting the efficacy of disentangled sensitive-irrelevant representations in enhancing fairness in downstream tasks. For example, the $\Delta_{EO}$ of DEFEND-D improves 1% and 12% than that of DEFEND-C and DEFEND on Reddit, respectively. **(4)** DEFEND-A underscores the effectiveness of the adversary in enhancing fairness, although this comes at a cost of accuracy. For instance, the improvement in AUC-ROC from 75.59% to 83.29% is accompanied by a considerable decrease in $\Delta_{DP}$ from 0.72% to 8.89%. In some cases, sacrificing partial accuracy to achieve higher fairness is a justified trade-off. **(5)** Compared to DEFEND+S, DEFEND not only achieves superior accuracy but also demonstrates enhanced fairness in most cases, which suggests that biased graph topology, negatively impacts the performance of reconstruct-based GAD method. This finding supports our choice to avoid structure reconstruction in the anomaly detection phase.

## 4.4 Parameter Analysis

In this section, we investigate the impact of three key parameters within DEFEND on Reddit, including $\alpha$ and $\gamma$ controlling the weight of prediction term and disentanglement term in disentangled representation learning, while $\beta$ controlling the weight of correlation constraints in the anomaly detection phase.

### 4.4.1 Impact of $\alpha$ and $\gamma$

To investigate the influence of predictiveness and disentanglement terms, We train DEFEND with the values of $\alpha$ and $\gamma$ among $\{0.1, 0.5, 1.0, 1.5, 2.0, 2.5, 5.0\}$. The results in terms of AUC-ROC and $\Delta_{EO}$ are presented in Figure 3 (a) and Figure 3 (b), respectively. Figure 3 (a) reveals that the anomaly detection efficacy remains substantially stable when $\alpha \geq 2.0$ and $\gamma \leq 1.0$, which represents an optimal range for anomaly detection capabilities. When $\gamma \geq 1.0$, there is a marked enhancement in fairness at the expense of a rapid decline

in accuracy. This trade-off occurs because larger $\gamma$ forces the model to achieve more similar predictions across demographic groups, potentially compromising its ability to identify true anomalies. Besides, a reduction in $\alpha$ alone significantly decreases both AUC-ROC and $\Delta_{EO}$. It underscores that without accurate sensitive attribute prediction, the model fails to effectively disentangle the representations, leading to both poor detection performance and unfair predictions. Therefore, selecting appropriate values for $\alpha$ and $\gamma$ is instrumental in navigating the tradeoff between anomaly detection accuracy and fairness.

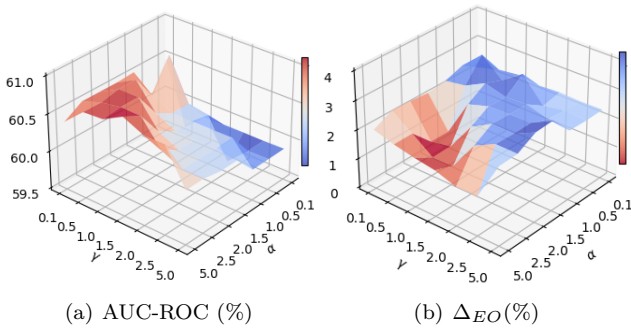

(a) AUC-ROC (%)          (b) $\Delta_{EO}$(%)

Figure 3: Impacts of varying predictiveness term weight $\alpha$ and disentanglement term weight $\gamma$ in Equation 15 on Reddit dataset in terms of AUC-ROC and $\Delta_{EO}$.

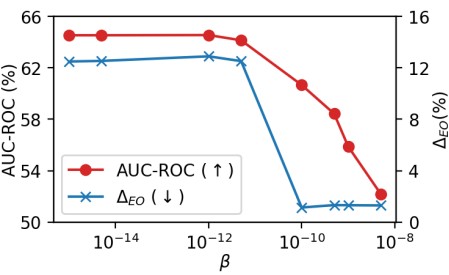

Figure 4: Impacts of varying correlation constraints weight $\beta$ in Equation 19 on Reddit.

### 4.4.2 Impact of $\beta$

To evaluate the effect of the constraint term, we further train DEFEND with different values of $\beta$ among $\{1e-15, 5e-15, 1e-12, 5e-12, 1e-10, 5e-10, 1e-9, 5e-9\}$ on Reddit. The results are depicted in Figure 4. First, we can observe that the $\Delta_{EO}$ decreases more than 10% while the AUC-ROC witnesses a marginal decline of nearly 4% at $\beta$ around $5e-12$. This favorable trade-off demonstrates that our constraint mechanism effectively reduces the model's reliance on sensitive information during anomaly scoring while largely preserving its detection capability. It emphasizes the necessity of integrating constraints within the reconstruction-based evaluation of anomalous nodes, considering the interrelation between input and sensitive attributes. Moreover, it is evident that both AUC-ROC and $\Delta_{EO}$ exhibit a decrement as the weight assigned to correlation constraints increases. It reveals that stronger constraints force the reconstruction process to be more independent of sensitive attributes, leading to fairer but potentially less discriminative representations. The judicious selection of $\beta$ is crucial for the trade-off between utility and fairness.

## 5 Related Work

### 5.1 Graph Anomaly Detection

Graph anomaly detection (GAD) has drawn rising interest since graph-structured data becoming increasingly prevalent in complex real-world systems. Given the absence of ground truth anomaly labels, many GAD methods focus on an unsupervised manner. Autoencoder is a prominent paradigm in this domain, which hypothesizes that the decoder cannot properly reconstruct anomalies deviating significantly from the majority. DOMINANT (Ding et al., 2019) employs GCN-based autoencoder and evaluates anomalies based on the reconstruction errors of node attributes and graph topology. AnomalyDAE (Fan et al., 2020) adopts a dual autoencoder structure to capture the cross-modality interactions between topology and node attributes. GAD-NR (Roy et al., 2023) further incorporates neighborhood reconstruction. Contrastive learning is another prevalent self-supervised paradigm in GAD. CoLA (Liu et al., 2021) firstly introduces node-subgraph contrast to identify anomalies based on the relation between nodes and their neighbors. Building upon this, ANEMONE (Jin et al., 2021) incorporates node-node contrast and GRADATE (Duan et al., 2023) incorporates subgraph-subgraph contrast to explore multi-level characteristics. Several studies also investigate anomaly detection across different levels of supervision (Chang et al., 2024; Xu et al., 2024; Liu et al., 2024;

2025). Despite the efficacy of these methods in anomaly detection, they are prone to biased decisions due to the neglect of sensitive attributes. To bridge this gap, we explore a novel problem of fair graph anomaly detection, aiming to detect anomalies impartially, without bias toward sensitive attributes.

## 5.2 Fairness on Graphs

Numerous studies have been conducted to mitigate source bias in training data to promote fairness in decision-making for graph learning tasks (Dong et al., 2023; Chen et al., 2023). Graph debiasing methods involve removing bias from the input graph before conducting target tasks. For example, FairWalk (Rahman et al., 2019) enhances the general random walk algorithm to capture more diverse neighborhoods, thereby producing embeddings that exhibit reduced bias, while FairDrop (Spinelli et al., 2021) alters graph topology to reduce homophily related to sensitive attributes. EDITS (Dong et al., 2022) goes further by adjusting both graph topology and node attributes based on the distance among demographic groups. In-processing methods represent another pipeline that revises the model training process to achieve more fair outcomes. For instance, FairGNN (Dai & Wang, 2021) integrates an adversary to achieve fair outputs for node classification with limited sensitive attributes, while Graphair (Ling et al., 2022) seeks to learn fair representations by automated graph data augmentations. FairVGNN (Wang et al., 2022) tackles the sensitive attribute leakage caused by feature propagation in GNNs by automatically learning from fair views. Additionally, FairGKD (Zhu et al., 2023) investigates the fairness performance in different training strategies and uses distilled knowledge from partial data training to enhance model fairness. However, directly applying the above methods for GAD poses significant challenges. A primary issue is the overlap between anomalous characteristics and sensitive attributes, which complicates the debiasing process. Besides, these methods are typically designed for or validated on supervised tasks, while the lack of ground truth is a fundamental issue in anomaly detection.

## 5.3 Fair Representation Learning

Fair representation learning has shown great success in learning representations free from sensitive information while maintaining downstream task-related information for decision-making (Zemel et al., 2013; Liu et al., 2022a). Disentangled representation learning provides a novel perspective in fair representation learning, enabling the simultaneous maintenance of sensitive-relevant and -irrelevant information, which are separated into independent subspaces. While disentangled fair representation learning has shown promise in benefiting fairness in image classification (Creager et al., 2019; Kim et al., 2021; Oh et al., 2022), its applicability on unsupervised GAD is non-trivial. The first challenge involves effectively encoding node attributes and graph topology into disentangled i.i.d. representations while preserving the information essential for anomaly detection. An additional challenge lies in achieving fair GAD based on sensitive-irrelevant representations. In prior studies, disentangled sensitive-irrelevant representations are directly used for supervised downstream tasks (Creager et al., 2019; Oh et al., 2022). However, in unsupervised GAD, where labels are not available, the reconstruction error serves as an effective criterion for decision-making. Considering the correlation between sensitive attributes and other attributes, as well as the connections influenced by sensitive attributes, direct reconstruction of the original graph may result in biased decisions.

## 6 Conclusion

In this paper, we propose DEFEND, a novel disentangle-based framework for fair graph anomaly detection, aiming to balance fairness and utility in decision-making. To the best of our knowledge, DEFEND is the first method in enhancing fairness in the task of graph anomaly detection. DEFEND first introduces disentangled representation learning to capture informativeness yet sensitive-irrelevant representations, thereby mitigating societal bias associated with sensitive attributes within the input graph. Furthermore, to alleviate discriminatory decisions in anomaly detection, DEFEND reconstructs input attributes from the sensitive-irrelevant representations and implements a constraint on the correlation between reconstruction error and predicted sensitive attributes. The effectiveness of DEFEND has been substantiated through extensive experiments on real-world datasets, demonstrating its superiority over several baselines regarding accuracy and fairness. We discuss limitations and future work in Appendix F.

**Acknowledgments**

This research is supported by a grant from the National Key Research and Development Program of China (No.2022YFF0711801) and the CAS 145 Informatization Project CAS-WX2022GC-0301. This work is supported in part by NSF under grants III-2106758 and POSE-2346158. Jianjun Yu is the corresponding author.

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

## A    Pseudo Code

The overall training process of DEFEND is presented in Algorithm 1. Firstly, given an attributed graph $\mathcal{G}$, we aim to train a variational graph autoencoder, in which the disentangled encoder $f_e$ can separate sensitive-irrelevant representations while maintaining information for reconstructing $\mathcal{G}$ by the decoder $f_a$ and $f_x$ (Line 4-6). We encourage the disentanglement with a learnable adversary $g_\omega$ optimized by $\mathcal{L}_{adv}$ (Line 8-9). The optimization of $f_e$, $f_a$ and $f_x$ and that of $g_\omega$ are conducted adversarially. Next, the frozen $f_e$ captures sensitive-irrelevant node representations (Line 14), and the decoder $f_\phi$ solely reconstructs attributes (Line 15). We utilize reconstruction error as the anomaly score and constrain the correlation between reconstruction error and predicted sensitive attributes (Lines 16-19). Finally, only $f_\phi$ will be updated by $\mathcal{L}_{ad}$ in the anomaly detection phase.

## B    Complexity Analysis

We analyze the time and space computational complexity of DEFEND. Let $N$, $E$, $d$ and $h$ denote the number of nodes, edges, attribute dimensions and hidden dimensions, respectively.

In the disentangled representation learning phase, the encoder $f_e$ exhibits a complexity of $O(Ed)$, while the decoders $f_a$ and $f_x$ exhibit a complexity of $O(N^2)$ for adjacency matrix reconstruction. The adversary $g_\omega$ operates at $O(Nh)$. For loss computations, the variational reconstruction loss $\mathcal{L}_{vae}$ exhibits $O(N^2)$ complexity for adjacency matrix reconstruction and $O(Nd)$ for attribute reconstruction. The disentanglement loss $\mathcal{L}_{dis}$, predictive loss $\mathcal{L}_{pre}$, and adversarial loss $\mathcal{L}_{adv}$ each exhibit $O(Nh)$ complexity. In the anomaly detection phase, the frozen encoder $f_e$ maintains $O(Ed)$ complexity, while the MLP-based decoder $f_\phi$ exhibits $O(Nh)$ complexity. Both the correlation constraint loss $\mathcal{L}_{corr}$ and reconstruction loss $\mathcal{L}_{rec}^X$ exhibit $O(Nh)$ complexity. Thus, the total time complexity is $O(N^2 + Ed)$. The space complexity of DEFEND is dominated by the storage of the dense adjacency matrix $O(N^2)$, node feature matrix $O(Nd)$, and model parameters with intermediate results $O(Nh)$, resulting in a total space complexity of $O(N^2 + Nd + Nh)$.

To enhance scalability, we implement a sparse version that computes reconstruction only for existing edges

---

**Algorithm 1** DEFEND algorithm

---

**Input:** Graph $\mathcal{G} = (\mathcal{V}, \mathcal{E})$, adjacency matrix $\mathbf{A}$, attribute matrix $\mathbf{X}$, sensitive attribute matrix $\mathbf{S}$, GNN encoder $f_e$, attribute decoder $f_x$, structure decoder $f_a$, binary adversary $g_\omega$, linear attribute decoder $f_\phi$, training iteration $T$;

**Output:** Anomaly scores $\mathbf{o}$

1:  *// Disentangled representation learning phase;*
2:  Initialize $f_e, f_a, f_x, g_\omega$;
3:  **while** not converged **do**
4:      $\mathbf{Z}_x, \mathbf{Z}_s \leftarrow f_e(\mathbf{X}, \mathbf{A})$;
5:      $\hat{\mathbf{A}}, \hat{\mathbf{X}} \leftarrow f_a(\mathbf{Z}_x, \mathbf{Z}_s), f_x(\mathbf{Z}_x, \mathbf{Z}_s)$;
6:      Calculate $\mathcal{L}_{total}$ according to Equation 15;
7:      Update $f_e, f_a, f_x$ by gradient descent using $\mathcal{L}_{total}$;
8:      Calculate $\mathcal{L}_{adv}$ according to Equation 16;
9:      Update $g_\omega$ by gradient descent using $\mathcal{L}_{adv}$;
10: **end while**
11: *// Anomaly detection phase;*
12: Freeze $f_e$ and initialize $f_\phi$;
13: **for** $t = 1, 2, \ldots, T$ **do**
14:     $\bar{\mathbf{Z}}_x, \mathbf{Z}_s \leftarrow f_e(\mathbf{X}, \mathbf{A})$;
15:     $\tilde{\mathbf{X}} \leftarrow f_\phi(\bar{\mathbf{Z}}_x, \tilde{\mathbf{Z}}_s)$;
16:     Calculate anomaly score $\mathbf{o}$ according to Equation 17;
17:     Calculate $\mathcal{L}_{ad}$ according to Equation 19;
18:     Update $f_\phi$ by gradient descent using $\mathcal{L}_{ad}$;
19: **end for**
20: **return** $\mathbf{o}$;

---

and approximates the global statistics using row/column means, instead of reconstructing the full adjacency matrix. We also replace dense matrix operations with sparse operations in loss computations. This optimization reduces the complexity of the decoder $f_a$ to $O(Eh)$ for structure reconstruction. Besides, the complexity of $\mathcal{L}_{vae}$ decreases to $O(E)$ for structure and maintains $O(Nd)$ for attribute reconstruction. Thus, the overall time complexity is reduced to $O(Ed + Nd)$. The space complexity decreases significantly to $O(E + Nd + Nh)$ by utilizing sparse matrix representations and eliminating dense adjacency matrices.

## C   Datasets

We employ three real-world datasets for fair GAD, which provide both real sensitive attributes and ground-truth labels for GAD. In Reddit and Twitter datasets, the sensitive attribute is the political leaning of users, while the anomaly label is assigned to misinformation spreaders. The Credit dataset focuses on payment default detection, with age as the sensitive attribute. Details of these datasets are summarized in Table 1.

- **Reddit** (Neo et al., 2024) contains 110 politically oriented subreddits, encompassing all historical postings within these forums. It also includes all historical posts from several active discussion participants. A relational graph was constructed by linking users who posted in the same subreddit within 24 hours.

- **Twitter** (Neo et al., 2024) is conducted on 47,712 users with historical posts, user profiles, and follower relationships. The user information, such as the organization status, was inferred using the M3 System from user profiles and tweets. The key account metrics and averaged post embeddings were combined to form node features. The network structure was established based on follower relationships between users.

- **Credit** (Agarwal et al., 2021) contains 30,000 individuals with features like education, credit history, age, and features derived from their spending and payment patterns. Two nodes are connected if their similarity exceeds 70% of the maximum similarity between all node pairs, measured using Minkowski distance.

## D   Baselines

In this subsection, we introduce the baselines employed in our experiments, including GAD methods (i.e., DOMINANT, CoLA, CONAD, and VGOD), Fairness Regularizers (i.e., FairOD, Correlation, and HIN), and Graph Debiasers(i.e., FairWalk and EDITS).

- **DOMINANT** (Ding et al., 2019) devises a GCN-based autoencoder to detect anomalies by reconstructing node attributes and graph structure. The attribute decoder is the reverse structure of the encoder and the structure decoder is applied by dot product. Anomalous nodes are evaluated by reconstruction errors.

- **CoLA** (Liu et al., 2021) is a contrastive self-supervised learning framework for anomaly detection. It conducts a node-subgraph contrast to capture anomalies that are dissimilar from their local neighbors. Anomalous nodes are evaluated by the agreement between each node and its neighboring subgraph with a GNN-based model.

- **CONAD** (Xu et al., 2022) introduces a contrastive learning framework that leverages human knowledge through data augmentation to enhance anomaly detection capabilities. It employs a Siamese graph neural network with a contrastive loss to encode both the modeled knowledge and the original attributed networks. Anomalous nodes are evaluated by reconstruction errors.

- **VGOD** (Huang et al., 2023) proposes a variance-based framework that combines a variance-based model for structural outlier detection with an attribute reconstruction model for contextual outlier detection. It achieves balanced detection performance between structural and contextual outliers while addressing data leakage issues present in existing injection-based approaches.

- **FairOD** (Shekhar et al., 2021) is a fairness-aware outlier detector on independent and identically distributed (i.i.d.) data. It devises a regularization term to prompt the fairness of demographic parity by minimizing the reconstruction errors $\mathbf{o}$ and the sensitive attributes $\mathbf{S}$.

$$\mathcal{L}_{DP}^{FairOD} = \left| \frac{\left( \sum_{i=1}^{n} o_i - \mu_o \right) \left( \sum_{i=1}^{n} s_i - \mu_s \right)}{\sigma_o \sigma_S} \right| \tag{20}$$

where $\mu_o$ and $\mu_s$ represent the means, while $\sigma_o$ and $\sigma_s$ denote the corresponding standard deviations of $\mathbf{o}$ and $\mathbf{S}$, respectively. Besides, it utilizes an approximation of Discounted Cumulative Gain (DCG) to enforce the group fidelity.

$$\mathcal{L}_{ADCG}^{FairOD} = \sum_{s \in \{0,1\}} \left( 1 - \sum_{\{v_i : s_i = s\}} \frac{2^{o_i^{base}} - 1}{\text{DNM}_s} \right), \tag{21}$$

where $\text{DNM}_s = \log_2 \left( 1 + \sum_{\{v_j : s_j = s\}} \sigma(o_j - o_i) \right)$ $\cdot$ $IDCG_s$ and $IDCG_s = \sum_j^{|\{v_j : s_j = s\}|} \left( 2^{o_j^{base}} - 1/\log_2(1 + j) \right)$. Here, $o_i^{base}$ is the reconstruction error of $v_i$ in the base model and $\sigma(\cdot)$ is the Sigmoid function. The overall loss of the model equipped with FairOD regularizer can be calculated by $\mathcal{L} = \mathcal{L}_{base} + \lambda \mathcal{L}_{DP}^{FairOD} + \gamma \mathcal{L}_{ADCG}^{FairOD}$, where $\lambda$ and $\gamma$ are weight parameters.

- **Correlation** (Shekhar et al., 2021) is an implementation of FairOD, which measures the correlation between sensitive attributes $\mathbf{S}$ and reconstruction errors $\mathbf{o}$ using the cosine similarity.

$$\mathcal{L}^{Corr} = \left| \frac{\mathbf{o} \cdot \mathbf{S}}{\sqrt{(\mathbf{o} \cdot \mathbf{o})(\mathbf{S} \cdot \mathbf{S})}} \right|, \tag{22}$$

where $(\cdot)$ represents the dot product of two vectors. The overall loss of the model equipped with the Correlation regularizer can be calculated by $\mathcal{L} = \mathcal{L}_{base} + \lambda \mathcal{L}^{Corr}$.

- **HIN** (Zeng et al., 2021) focuses on fair representation learning for heterogeneous information networks. The demographic parity-based fairness-aware loss function is calculated by:

$$\mathcal{L}_{DP}^{HIN} = \sum_{k \in \{0,1\}} \left( \frac{\sum_{\{v_i : s_i = 1\}} P(\hat{y}_i = k)}{|\{v_i : s_i = 1\}|} - \frac{\sum_{\{v_i : s_i = 0\}} P(\hat{y}_i = k)}{|\{v_i : s_i = 0\}|} \right)^2, \tag{23}$$

where $P(\hat{y}_i = 1)$ denotes the predicted probability of $v_i$ to be identified as an anomalous node. The equal opportunity-based fairness-aware loss is calculated by:

$$\mathcal{L}_{EO}^{HIN} = \sum_{k \in \{0,1\}} \left( \frac{\sum_{\{v_i : s_i = 1, y = k\}} Pr(\hat{y}_i = k)}{|\{v_i : s_i = 1, y = k\}|} - \frac{\sum_{\{v_i : s_i = 0, y = k\}} Pr(\hat{y}_i = k)}{|\{v_i : s_i = 0, y = k\}|} \right)^2. \tag{24}$$

As the calculation of $\mathcal{L}_{EO}^{HIN}$ requires task-related labels and anomaly detection tasks typically lack ground truth labels, we use $\mathcal{L}_{ADCG}^{FairOD}$ as a replacement to enhance the fairness in equal opportunity. Thus, the modified overall loss of the model equipped with HIN regularizer can be calculated by:

$$\mathcal{L} = \mathcal{L}_{base} + \lambda \mathcal{L}_{DP}^{HIN} + \gamma \mathcal{L}_{ADCG}^{FairOD}. \tag{25}$$

- **FairWalk** (Rahman et al., 2019) introduces a fairness-aware embedding method that generates node embeddings by considering sensitive attributes and the topology of the graph. It enhances the general random walk algorithm to capture more diverse neighborhoods, thereby producing embeddings that exhibit reduced bias.

- **EDITS** (Dong et al., 2022) is a graph debiasing method for attributed graphs, mitigating bias present in both graph topology and node features. It minimizes the approximated Wasserstein distance between the distributions of different groups for any attribute dimension to enhance group fairness.

## E    Implementation Details

For DOMINANT, CoLA, CONAD and VGOD, we use the code and default hyper-parameters provided by PyGOD[1] (Liu et al., 2022c). For FairWalk[2] and EDITS[3], we implement them using the code published by their authors. For FairOD, HIN, and Correlation, we implement the code provided by FairGAD[4].

## F    Limitations and Future Work

First, DEFEND encounters difficulties when protected attributes are highly correlated with other features, potentially resulting in information loss during disentanglement. Additionally, DEFEND relies on adversarial learning to approximate the Total Correlation penalty used for disentanglement. This dependency introduces potential convergence instability issues that might compromise the robustness of learned representations (Oh et al., 2022). We plan to extend the framework to handle multiple and continuous sensitive attributes simultaneously, which would enhance its practical applicability.

---

[1]https://pygod.org
[2]https://github.com/urielsinger/fairwalk
[3]https://github.com/yushundong/EDITS
[4]https://github.com/nigelnnk/FairGAD

