# OpenReview forum: "Enhancing Fairness in Unsupervised Graph Anomaly Detection through Disentanglement"
_TMLR — Accepted by TMLR_

### Review · Reviewer_niAE · 2024-12-15

**Summary Of Contributions:**

This paper proposes DEFEND, a framework for fair graph anomaly detection (GAD) that reduces bias by learning sensitive-irrelevant representations and using attribute-focused reconstruction to enhance both fairness and utility.

**Audience:**

Yes

**Claims And Evidence:**

Yes

**Requested Changes:**

Besides addressing the above weaknesses, here are some suggestions:
1. Add a pseudo code of the whole algorithm for better understanding.
2. Please add a computational analysis of the proposed method.

**Strengths And Weaknesses:**

Strengths:
1. Considering fairness in unsupervised graph anomaly detection is quite interesting and holds practical significance.
2. The method is clear and easy to follow.

Weakness:
1. The organization of the paper needs improvement.
   - In particular, the Introduction lacks a clear, logical flow, resulting in an unconvincing presentation of the shortcomings of existing methods for incorporating fairness into graph anomaly detection and the motivations for the proposed approach.
   - The method follows an encoder-decoder framework, but Fig. 1 does not clearly reflect this structure. It should be revised to better align with and illustrate the proposed method.

2. The paper relies on the key assumption that the latent space can be decomposed into two independent subspaces. However, it does not provide sufficient theoretical analysis or detailed justification for the independence between sensitive-irrelevant and sensitive-relevant representations, which, in my opinion, are likely to be highly dependent. This assumption should be addressed and substantiated more carefully.

3. In the experiments, while the authors explore different hyperparameters, they fail to provide deeper insights into how their method works, limiting the understanding of its underlying mechanisms.

---

> ### Author Response · Authors · 2024-12-31
> **Response to Reviewer niAE (Part 1/2)**
>
> **Revise the introduction to better clarify the motivation.**
>
> We appreciate this feedback. We have revised our introduction to clarify our motivations. We now cite previous studies to demonstrate that existing GAD methods have paid insufficient attention to fairness concerns. We then summarize the key challenges for fair GAD, including the fundamental challenge of absent labels and the graph bias existing in both graph topology and node attributes. Finally, we clarify the motivation behind each component of our design, showing how they address these challenges and achieve a balance between fairness-utility trade-off. We hope the revised introduction provides a clearer view for our method.
>
> **Improve framework figure to better illustrate the architecture of DEFEND.**
>
> We have revised Figure 1 to better illustrate the encoder-decoder architecture of our method. We welcome any additional suggestions for further improving the visualization.
>
> **Theoretical analysis of the independence and visualization between sensitive-irrelevant and sensitive-relevant representations**
>
> To ensure the disentanglement of sensitive-irrelevant representations $\mathbf{Z} _ x$ and sensitive-relevant representations $\mathbf{Z} _ s$, it is imperative that the aggregate posterior distribution can be factorized as $q(\mathbf{Z}_x,\mathbf{Z}_s)=q(\mathbf{Z}_x)q(\mathbf{Z}_s)$. We employ the KL divergence between $q(\mathbf{Z}_x,\mathbf{Z}_s)$ and $q(\mathbf{Z}_x)q(\mathbf{Z}_s)$ as the disentanglement criteria, formulated as ${KL}\big[q(\mathbf{Z}_x, \mathbf{Z}_s) \Vert q(\mathbf{Z}_x)q(\mathbf{Z}_s)\big]$, when KL=0 means the independence between $\mathbf{Z}_x$ and $\mathbf{Z}_s$. Thus, we aim to minimize the ${KL}\big[q(\mathbf{Z}_x, \mathbf{Z}_s)  \Vert q(\mathbf{Z}_x)q(\mathbf{Z}_s)\big]$to ensure the disentanglement.
>
> Following previous studies [1,2,3], the availability of samples from both distributions allows us to minimize the KL divergence using the density-ratio trick, which involves training a classifier to approximate the density ratio inherent in the KL term, i.e.,
>
> $$
>             {KL}\big[q(\mathbf{Z} _ x, \mathbf{Z} _ s) \Vert q(\mathbf{Z}_x)q(\mathbf{Z} _ s)\big]=\mathbb{E} _ {q(\mathbf{Z} _ x, \mathbf{Z} _ s)} \log \frac{q(\mathbf{Z} _ {x},\mathbf{Z} _ {s})}{q(\mathbf{Z} _ {x})q(\mathbf{Z} _ {s})} \\ \approx \mathbb{E} _ {q(\mathbf{Z} _ x, \mathbf{Z} _ s)} [\log g _ \omega(\tilde{\mathbf{y}}=1|\mathbf{Z} _ x, \mathbf{Z} _ s ) - \log g _ \omega(\tilde{\mathbf{y}}=0|\mathbf{Z} _ x, \mathbf{Z} _ s)],
> $$
>
> where $g_{\omega}$ is the MLP-based binary classifier serving as an adversary in DEFEND, and $g _ \omega(\tilde{\mathbf{y}}=1|\mathbf{Z} _ x, \mathbf{Z} _ s)$ is the probability that the samples are from $q(\mathbf{Z} _ x,\mathbf{Z} _ s)$ and $g _ \omega(\tilde{\mathbf{y}}=0|\mathbf{Z} _ x, \mathbf{Z} _ s)$ is the probability that the samples are from $q(\mathbf{Z} _ x)q(\mathbf{Z} _ s)$.  Thus, we can achieve disentanglement by minimizing $\mathcal{L} _ {dis}=\mathbb{E} _ {q(\mathbf{Z} _ x, \mathbf{Z}_ s)} [\log g _ \omega(\tilde{\mathbf{y}}=1|\mathbf{Z} _ x, \mathbf{Z} _ s) - \log g _ \omega(\tilde{\mathbf{y}}=0|\mathbf{Z} _ x, \mathbf{Z} _ s)]$.
>
> We further provide visualizations of sensitive-irrelevant representations Z_x and sensitive-relevant representations Z_s in [this figure](https://i.postimg.cc/htR1ZGGq/image.png). In the left figure, Z_x is dimensionally reduced by t-SNE and colored by sensitive attribute. We can observe that the points with different sensitive attributes (s=0 and s=1) are well-mixed together, indicating that Z_x successfully maintains sensitive-irrelevant information. In the right figure, we fit Gaussian distributions to Z_s across different sensitive attribute groups. We can observe that while there is considerable overlap, the distributions of Z_s show different central tendencies for s=0 and s=1, with s=1 group slightly shifted towards more positive values. These visualizations demonstrate that our disentanglement approach achieves somewhat separation between sensitive-relevant and sensitive-irrelevant information, even though the separation is not perfect.
>
> **Enhance hyperparameter analysis with deeper insights.**
>
> We appreciate this feedback. In the revised manuscript, we have enhanced our analysis with deeper insights into how DEFEND works.

---

> ### Author Response · Authors · 2024-12-31
> **Response to Reviewer niAE (Part 2/2)**
>
> **Pseudocode and Computational Analysis.**
>
> We provide a detailed pseudo code of the algorithm in Appendix C to clarify the implementation details, and a comprehensive computational complexity analysis in Appendix D. Here we present DEFEND's algorithm procedure and computational complexity:
>
>  **Algorithm Workflow**. DEFEND operates in two phase: (1) **Disentangled Representation Learning**. Given an attributed graph $\mathcal{G}$, we aim to train a variational graph autoencoder, in which the disentangled encoder $f_{e}$ can separate sensitive-irrelevant representations while maintaining information for reconstructing $\mathcal{G}$ by the decoder $f_{a}$ and $f_{x}$ (Line 4-6). We encourage the disentanglement with a learnable adversary $g_{\omega}$ optimized by $\mathcal{L}{adv}$ (Line 8-9). The optimization of $f{e}$, $f_{a}$ and $f_{x}$ and that of $g_{\omega}$ are conducted adversarially. (2) **Anomaly Detection**.  Next, the frozen $f_{e}$ captures sensitive-irrelevant node representations (Line 14), and the decoder $f_{\phi}$ solely reconstructs attributes (Line 15). We utilize reconstruction error as the anomaly score and constrain the correlation between reconstruction error and predicted sensitive attributes (Lines 16-19). Finally, only $f_{\phi}$ will be updated by $\mathcal{L}_{ad}$ in the anomaly detection phase.
>
> **Complexity Analysis**. We analyze the time and space computational complexity of \method. Let $N$, $E$, $d$ and $h$ denote the number of nodes, edges, attribute dimensions and hidden dimensions, respectively.
> In the disentangled representation learning phase, the encoder $f_e$ exhibits a complexity of $O(Ed)$, while the decoders $f_a$ and $f_x$ exhibit a complexity of $O(N^2)$ for adjacency matrix reconstruction. The adversary $g_\omega$ operates at $O(Nh)$. For loss computations, the variational reconstruction loss $\mathcal{L}{vae}$ exhibits $O(N^2)$ complexity for adjacency matrix reconstruction and $O(Nd)$ for attribute reconstruction. The disentanglement loss $\mathcal{L}{dis}$, predictive loss $\mathcal{L}{pre}$, and adversarial loss $\mathcal{L}{adv}$ each exhibit $O(Nh)$ complexity. In the anomaly detection phase, the frozen encoder $f_e$ maintains $O(Ed)$ complexity, while the MLP-based decoder $f\phi$ exhibits $O(Nh)$ complexity. Both the correlation constraint loss $\mathcal{L}{corr}$ and reconstruction loss $\mathcal{L}{rec}^X$ exhibit $O(Nh)$ complexity. Thus, the total time complexity is $O(N^2 + Ed))$. The space complexity of \method is dominated by the storage of the dense adjacency matrix $O(N^2)$, node feature matrix $O(Nd)$, and model parameters with intermediate results $O(Nh)$, resulting in a total space complexity of $O(N^2 + Nd + Nh)$.
>
> To enhance scalability, we implement a sparse version that computes reconstruction only for existing edges and approximates the global statistics using row/column means, instead of reconstructing the full adjacency matrix. We also replace dense matrix operations with sparse operations in loss computations. This optimization reduces the complexity of the decoder $f_a$ to $O(Eh)$ for structure reconstruction. Besides, the complexity of $\mathcal{L}_{vae}$ decreases to $O(E)$ for structure and maintains $O(Nd)$ for attribute reconstruction. Thus, the overall time complexity is reduced to $O(Ed + Nd)$. The space complexity decreases significantly to $O(E + Nd + Nh)$ by utilizing sparse matrix representations and eliminating dense adjacency matrices.

---

### Review · Reviewer_pyYs · 2024-12-15

**Summary Of Contributions:**

The authors introduce DEFEND, a GNN-based anomaly detection framework using disentangelement and reconstruction-based objectives to fairly detect anomolous nodes. DEFEND is based on the observation that sensitive attributes are often associated with other sensitive and sensitive-irrelevant as well as benign attributes, making fair training difficult. To combat, this, DEFEND disentangles node representations in order to separate sensitive-relevant and sensitive-irrelevant features which in turn benefits fair graph anomaly detection.

**Audience:**

Yes

**Claims And Evidence:**

Yes

**Requested Changes:**

Please address the above weaknesses. Also, the reconstruction loss and KL loss in Equations 11 and 12 as well as labeling the variational lower bound term as $\L_{vae}$ seems greatly inspired with the variational autoencoder from the popular [1], but it is not cited.

[1] Kingma and Welling. Auto-Encoding Variational Bayes. ICLR 2014.

**Strengths And Weaknesses:**

**Strengths**

- The authors conduct extensive experiments on 2 datasets with a very comprehensive overview of competing methods and frameworks.
- This work is transparent about the tradeoff between performance and fairness, and it strongly argues for the overall optimality of DEFEND given this context.
- Using disentanglement to combat entangled features for fairness is a fairly straightforward and intuitive idea, and it is good to see that it shows promise.

**Weaknesses**

- DEFEND achieves its effectiveness through several loss functions. While the authors delve into time complexity of the backbone architecture in Appendix C, the efficiency impact of each of thse loss terms is unclear.
- DEFEND claims to effectively disentangle sensitive-relevant and sensitive-relevant attributes, but there is no feature analysis to quantiatively assess its disentanglement abilities.
- The baseline methods are not based on disentanglement. DEFEND would be stronger if it also demonstrates superiority over some basic disentangelement-based methods.
- DEFEND+S exhibits worse performance overall when including a graph structure decoder. However, a method like EDITS could help with debiasing the graph itself before DEFEND is applied. Have the authors tried a variety of methods with eliminating graph structure bias?
- DEFEND acknowledges the existence of bias in graph topology, but no module seems tailored to combat this. In fact, its encoder backbone is a GCN, which leverages the graph topology.

---

> ### Author Response · Authors · 2024-12-31
> **Response to Reviewer pyYs**
>
> **Efficiency analysis of loss terms.**
>
> We provide an analysis of the computational complexity for each loss term in DEFEND. Let $N$ denotes the number of nodes, $E$ the number of edges, $d$ the attribute dimension, and $h$ the hidden dimension.
>
> In the disentangled representation learning phase, the variational reconstruction loss $\mathcal{L} _ {vae}$ exhibits $O(N^2)$ complexity for structure reconstruction and $O(Nd)$ for attribute reconstruction. Through our sparse implementation that computes reconstruction only for existing edges and employs sparse matrix operations, the complexity of $\mathcal{L} _ {vae}$ can be effectively reduced to $O(E)$ for structure reconstruction while maintaining $O(Nd)$ for attribute reconstruction. The disentanglement loss $\mathcal{L} _ {dis}$, predictive loss $\mathcal{L} _ {pre}$, and adversarial loss $\mathcal{L}_ {adv}$ each exhibit $O(Nh)$ complexity.
>
> In the anomaly detection phase, both the correlation constraint loss $\mathcal{L} _ {corr}$ and reconstruction loss $\mathcal{L} _ {rec}^X$ exhibit $O(Nh)$ complexity.
>
> The loss terms do not significantly affect the overall time complexity. Detailed computational analysis are provided in Appendix D.
>
> **Visualizations of disentangled representations.**
>
> We provide visualizations of sensitive-irrelevant representations Z_x and sensitive-relevant representations Z_s in [this figure](https://i.postimg.cc/htR1ZGGq/image.png). In the left figure, Z_x is dimensionally reduced by t-SNE and colored by sensitive attribute. We can observe that the points with different sensitive attributes (s=0 and s=1) are well-mixed together, indicating that Z_x successfully maintains sensitive-irrelevant information. In the right figure, we fit Gaussian distributions to Z_s across different sensitive attribute groups. We can observe that while there is considerable overlap, the distributions of Z_s show different central tendencies for s=0 and s=1, with s=1 group slightly shifted towards more positive values. These visualizations demonstrate that our disentanglement approach achieves somewhat separation between sensitive-relevant and sensitive-irrelevant information, even though the separation is not perfect.
>
> **Compare with disentangelement-based methods.**
>
> We appreciate this valuable suggestion. We want to clarify that DEFEND is the first approach that leverages disentangled representation learning for fair graph anomaly detection. While there are no directly comparable disentanglement-based methods in this specific domain, we welcome any suggestions for relevant baselines that could strengthen our experimental evaluation.
>
> **Enhance DEFEND+S with debiased graphs.**
>
> We conducted experiments using EDITS to debias the graph on the Twitter before applying DEFEND+S. We have the following observations. In terms of fairness metrics, EDITS+DEFEND+S achieves better \Delta_{EO} compared to DEFEND+S, however, the \Delta_{DP} becomes worse than DEFEND+S. Regarding detection performance, EDITS+DEFEND+S shows decreased performance in both AUC-ROC and AUC-PR. This performance degradation aligns with our observations for some other GAD methods (as shown in Tables 2 and 4). These results indicate that while graph debiasing can help with certain fairness metrics, it may compromise the model's ability to detect anomalies effectively, highlighting the challenge of balancing fairness and detection performance in graph anomaly detection.
>
> |  | AUC-ROC | AUC-PR | \Delta_{DP} | \Delta_{EO} |
> | --- | --- | --- | --- | --- |
> | EDITS+DEFEND+S | 48.27$\pm$1.73 | 6.10$\pm$0.24 | 2.20$\pm$0.45 | 0.84$\pm$0.55 |
> | DEFEND+S | 50.15$\pm$0.74 | 6.48$\pm$0.13 | 1.28$\pm$0.19 | 1.59$\pm$1.21 |
> | DEFEND | 75.59$\pm$2.00 | 11.85$\pm$0.79 | 0.72$\pm$0.70 | 0.79$\pm$0.61 |
>
> **Rationale for GCN-based Encoder.**
>
> While DEFEND uses GCN as the encoder backbone, we explicitly address topology bias through the disentanglement representation learning. The disentangled graph encoder separates sensitive-relevant and sensitive-irrelevant representations, allowing us to identify and mitigate biases in both graph topology and node attributes. This approach enables us to effectively handle bias in graph topology while preserving useful structural information into sensitive-irrelevant representations.
>
> **Add citation of the variational autoencoder.**
>
> Thank you for pointing this out. We have now added the citation in the revised manuscript.

---

> > ### Comment · Reviewer_pyYs · 2025-01-09
> > **Thank you for the response**
> >
> > **Efficiency**
> >
> > Thank you for including this discussion and for making your efficiency-saving implementation clear. I'm satisfied with this.
> >
> > **Visualization**
> >
> > Thank you for these visualizations; they are helpful. I agree that the distributions for $Z_s$ for $s=0$ and $s=1$ bear considerable overlap. However, it is difficult to evaluate this without a baseline for disentanglement, e.g. how well-separated are sensitive-relevant and sensitive-irrelevant attributes in competing baselines.
> >
> > **Comparison with disentanglement methods**
> >
> > I think a very simple $\beta$-VAE adapted to graphs with $\beta > 1$ could be a good nominal baseline. If this performs better than the most naive methods in the literature, I think that's enough to show the effectiveness of disentanglement as well as the merits of the more mindful approach to disentanglement that DEFEND takes. However, I recognize that this response is a bit late, so I don't consider the inclusion of this experiment to be very critical.
> >
> > **Using debiased graphs**
> >
> > Thank you for providing these experiments. I am satisfied with this response, and I think it also answers my final concern.
> >
> > **Rationale for GCN encoder**
> >
> > I think my main point is that all of DEFEND's loss functions are primarily feature-based and are concerned with disentangling sensitive attributes. While GCN incorporates some topological bias in its produced embeddings due to message-passing, DEFEND's modules do not take the graph structure explicitly into account. That said, your experiments on DEFEND+S with debiased graphs is helpful as it seems DEFEND does benefit from methods which directly consider topology in at least one metric, while other metrics suffer. This largely resolves my concern.

---

> > > ### Author Response · Authors · 2025-01-11
> > >
> > > Thank you for this insightful suggestion. We implemented $\beta$-VAE ($\beta \in {1.5, 2.0, 2.5}$) and selected the dimension with highest correlation to sensitive attributes as $Z_s$, with remaining dimensions as sensitive-irrelevant representations for downstream anomaly detection. The results are provided as follows.
> > >
> > > |  | AUC-ROC $\uparrow$ | AUC-PR $\uparrow$ | $\Delta_{DP} \downarrow$ | $\Delta_{EO} \downarrow$ |
> > > | --- | --- | --- | --- | --- |
> > > | DOMINANT | 60.82$\pm$0.09 | 20.02$\pm$0.04 | 13.20$\pm$0.08 | 5.59$\pm$0.19 |
> > > | CoLA | 45.20$\pm$0.98 | 17.90$\pm$1.54 | 4.95$\pm$1.78 | 4.06$\pm$1.85 |
> > > | CONAD | 60.81$\pm$0.10 | 20.02$\pm$0.05 | 13.32$\pm$0.33 | 5.70$\pm$0.37 |
> > > | VGOD | 72.01$\pm$0.93 | 39.38$\pm$2.45 | 42.47$\pm$5.61 | 46.79$\pm$6.09 |
> > > | $\beta$-VAE | 59.10$\pm$1.46 | 16.18$\pm$0.66 | 3.10$\pm$1.59 | 3.03$\pm$2.08 |
> > > | DEFEND| 60.67$\pm$0.42 | 16.46$\pm$0.25 | 0.94$\pm$0.72 | 1.13$\pm$0.81 |
> > >
> > > While $\beta$-VAE indeed outperforms basic GAD methods, it shows limitations compared to DEFEND. We argue this is because  $\beta$-VAE performs unsupervised disentanglement without knowledge of sensitive attributes - it aims to discover independent factors rather than explicitly separating sensitive information as defined.
> > >
> > > We provide visualizations of sensitive-irrelevant representations $Z_x$ and sensitive-relevant representations $Z_s$ of $\beta$-VAE in [this figure](https://i.postimg.cc/JzHbV4MK/reddit-0-5-1-5-2000-1-emb-tsne.png). Comparing with the disentanglement of DEFEND in [this figure](https://i.postimg.cc/htR1ZGGq/image.png), we observe that DEFEND demonstrates superior disentanglement performance as evidenced by its more uniform mixing between sensitive groups in $Z_x$ and clearer separation between groups with distinct peaks in $Z_s$.

---

> > > > ### Comment · Reviewer_pyYs · 2025-01-12
> > > > **Response**
> > > >
> > > > Thank you for the extra experiments; I am convinced of DEFEND's effectiveness. Comparison of the latent embeddings of both $\beta$-VAE and DEFEND are also helpful. My concerns are resolved.

---

### Review · Reviewer_6KZ6 · 2024-12-18

**Summary Of Contributions:**

This paper addresses the critical and underexplored issue of fairness in graph anomaly detection (GAD), proposing a comprehensive framework, DEFEND, to mitigate bias while maintaining strong anomaly detection capabilities. The key contributions include:

- Theoretically grounded disentangled representation learning for graphs
- Reconstruction-based anomaly detection that intentionally avoids biased graph topology
- Novel correlation constraints between reconstruction errors and sensitive attributes
- Computationally efficient implementation with strategies for reducing matrix reconstruction complexity

**Audience:**

Yes

**Broader Impact Concerns:**

NAN.

**Claims And Evidence:**

Yes

**Requested Changes:**

**Critical Changes**
1. Scalability Analysis: Provide runtime and scalability experiments on larger or denser graphs to assess DEFEND’s computational feasibility.
2. Hyperparameter Selection: Elaborate on the hyperparameter tuning process and its influence on performance, particularly for fairness-related terms.
3. Discussion of Limitations: Include a section addressing potential failure modes or limitations, such as scenarios where disentanglement might not be effective.

**Suggested Improvements**

1. Expand Fairness Metrics: Consider evaluating DEFEND using additional fairness criteria, such as individual fairness or measures specific to subgroup disparities.
2. Broaden Dataset Scope: If feasible, extend experiments to include more diverse datasets with varying graph structures and sensitive attributes.
3. Visual Representation: Include visualizations of disentangled representations to better demonstrate how sensitive-relevant and sensitive-irrelevant information is separated.
4. Qualitative Examples: Provide qualitative insights into specific cases where DEFEND outperforms baselines in terms of fairness.
5. Algorithmic Transparency: Add pseudocode for key components of the DEFEND framework to enhance reproducibility.
6. Detailed Discussion of Extensions: Offer more concrete insights into adapting DEFEND for multiple or continuous sensitive attributes, potentially with experimental validation.

**Strengths And Weaknesses:**

**Strengths**
1. Novelty: DEFEND introduces a first-of-its-kind framework for fair unsupervised GAD, combining ideas from disentangled representation learning and fairness-aware anomaly detection.
2. Theoretical Rigor: The method is built on a strong mathematical foundation, with detailed formulations for disentanglement and fairness constraints.
3. Comprehensive Evaluation: The paper provides extensive empirical results, including ablation studies and parameter sensitivity analysis, highlighting the importance of each component in DEFEND.
4. Practical Relevance: The proposed method is relevant for high-stakes applications like fraud detection, where fairness is critical.
5. Discussion of Generalization: The paper includes a thoughtful discussion on extending DEFEND to handle more complex sensitive attributes, such as categorical or continuous variables.

**Weaknesses**

1. Dataset Limitation: The experimental evaluation is conducted on only two real-world datasets (Reddit and Twitter), limiting the generalizability of findings across diverse graph structures.
2. Baseline Comparisons: Some baselines are adaptations of methods not originally designed for fair GAD, which could lead to less direct comparisons.
3. Efficiency Analysis: Limited discussion on computational efficiency and scalability to larger graphs or denser datasets.
4. Fairness Metrics: The evaluation focuses on demographic parity and equal opportunity but does not consider other fairness measures, such as individual fairness or subgroup-specific biases.
5. Lack of Qualitative Insights: The paper does not include qualitative analyses of specific cases where DEFEND improves fairness, which would help interpretability.

---

> ### Author Response · Authors · 2024-12-31
> **Response to Reviewer 6KZ6 (Part 1/3)**
>
> **Compare with methods specifically designed for fair GAD.**
>
> To the best of our knowledge, DEFEND is the first dedicated method for fair unsupervised graph anomaly detection. Given that existing GAD methods rarely consider fairness, we adapt established fairness methods from both anomaly detection and graph domains to enhance fairness in GAD. (1) Fairness regularizers. FairOD and Correlation are designed for i.i.d. anomaly detection, while HIN is a fairness approach designed for heterogeneous graphs. (2) Graph debiasers. EDITS and FairWalk are specifically designed to address bias in graph data. The details of baseline methods and their implementations are provided in  Appendix B and Appendix E.
>
> **Scalability Analysis.**
>
> We conducted comprehensive scalability analyses from both theoretical and empirical perspectives to demonstrate DEFEND's computational feasibility. For theoretical analysis, DEFEND achieves time complexity of $O(Ed + Nd)$ and space complexity of $O(Nd + E)$ through our sparse implementation, where $N$ is the number of nodes, $E$ is the number of edges, and $d$ is the attribute dimension. The detailed computational analysis is provided in Appendix D. Empirically, we conducted runtime and peak GPU memory utilization analyses comparing DEFEND with DOMINANT and its variants that require no preprocessing for graph debiasing. We measured the averaged epoch time across 10 runs for each method. Results demonstrate that DEFEND achieves superior performance in both runtime and memory utilization than DOMINANT and its variants.
> | Method | Model | Reddit |  | Twitter |  | Credit |  |
> |--------|--------|---------|---------|-----------|---------|---------|---------|
> | | | Memory (MB) | Runtime (s) | Memory (MB) | Runtime (s) | Memory (MB) | Runtime (s) |
> |-|DOMINANT|8848|0.123|58154|2.932|22974|0.555|
> |FairOD|DOMINANT|8848|0.242|58156|5.945|22976|1.131|
> |HIN|DOMINANT|8848|0.247|58156|6.006|22976|1.136|
> |Corr.|DOMINANT|8848|0.122|58154|3.004|22974|0.558|
> |Ours|DEFEND|7974|0.074|49570|0.176|19984|0.08|
>
> **Hyperparameter Selection.**
>
> We conducted hyperparameter tuning as follows: For the basic architecture parameters (hidden dimension, learning rate, and training epochs), we selected values based on model convergence and common practices in graph neural networks. For the loss weight coefficients $\alpha$, $\beta$, and $\gamma$ that balance different objectives, we explored their combinations to generate accuracy-fairness trade-off curves. The final parameters were selected by choosing points near the upper-left corner of these curves, which represent a good balance between anomaly detection performance and fairness metrics. More implementation details are provided in Section 4.1. We acknowledge that hyperparameter selection for unsupervised algorithms is non-trivial when applying them to new scenarios, and it remains an active research direction in the community [1,2].
>
> [1] Aggarwal, Charu C. "Ensembles for Outlier Detection and Evaluation." In *Proceedings of the 33rd ACM International Conference on Information and Knowledge Management*, pp. 1-1. 2024.
>
> [2] Ma, Martin Q., Yue Zhao, Xiaorong Zhang, and Leman Akoglu. "The need for unsupervised outlier model selection: A review and evaluation of internal evaluation strategies." *ACM SIGKDD Explorations Newsletter* 25, no. 1 (2023): 19-35.
>
> **Discussion of Limitations.**
>
> Following your suggestion, we have added a discussion of DEFEND's limitations in Appendix F. DEFEND encounters difficulties when protected attributes are highly correlated with other features, potentially resulting in information loss during disentanglement. Additionally, DEFEND relies on adversarial learning to approximate the Total Correlation penalty used for disentanglement. This dependency introduces potential convergence instability issues that might compromise the robustness of learned representations [1]. We plan to extend the framework to handle multiple and continuous sensitive attributes simultaneously would enhance its practical applicability.
>
> [1] Oh, Changdae, Heeji Won, Junhyuk So, Taero Kim, Yewon Kim, Hosik Choi, and Kyungwoo Song. "Learning fair representation via distributional contrastive disentanglement." In *Proceedings of the 28th ACM SIGKDD Conference on Knowledge Discovery and Data Mining*, pp. 1295-1305. 2022.

---

> ### Author Response · Authors · 2024-12-31
> **Response to Reviewer 6KZ6 (Part 2/3)**
>
> **Expand Fairness Metrics.**
>
> Thanks for your suggestion. Our study focuses on enhancing group fairness in graph anomaly detection. Demographic parity and equal opportunity are two widely adopted fairness metrics in previous studies [1, 2, 3, 4, 5]. While individual fairness is also important, it is beyond our scope in this work. Instead, we focus on group fairness in this work due to a critical challenge in anomaly detection: the inherent tension between statistical minority detection and societal minority protection. Anomaly detectors are designed to identify statistical minority samples, which can inadvertently lead to bias against societal minorities (defined by race/ethnicity/sex/age) due to their smaller group sizes. This becomes problematic when minority status (e.g., Hispanic) does not correlate with actual anomalous behavior (e.g., fraud). Without fairness considerations, such bias can result in disparate impact through overpolicing of minority groups and, consequently, underpolicing of majority groups.
>
> [1] Ling, Hongyi, Zhimeng Jiang, Youzhi Luo, Shuiwang Ji, and Na Zou. "Learning fair graph representations via automated data augmentations." In *International Conference on Learning Representations (ICLR)*. 2023.
>
> [2] Zhu, Yuchang, Jintang Li, Liang Chen, and Zibin Zheng. "The Devil is in the Data: Learning Fair Graph Neural Networks via Partial Knowledge Distillation." In *Proceedings of the 17th ACM International Conference on Web Search and Data Mining*, pp. 1012-1021. 2024.
>
> [3] Dong, Yushun, Ninghao Liu, Brian Jalaian, and Jundong Li. "Edits: Modeling and mitigating data bias for graph neural networks." In *Proceedings of the ACM web conference 2022*, pp. 1259-1269. 2022.
>
> [4] Dai, Enyan, and Suhang Wang. "Say no to the discrimination: Learning fair graph neural networks with limited sensitive attribute information." In *Proceedings of the 14th ACM International Conference on Web Search and Data Mining*, pp. 680-688. 2021.
>
> [5] Neo, Neng Kai Nigel, Yeon-Chang Lee, Yiqiao Jin, Sang-Wook Kim, and Srijan Kumar. "Towards Fair Graph Anomaly Detection: Problem, Benchmark Datasets, and Evaluation." In *Proceedings of the 33rd ACM International Conference on Information and Knowledge Management*, pp. 1752-1762. 2024.
>
> **Broaden Dataset Scope.**
>
> We further evaluated our method on the Credit dataset, which provides a distinct application scenario from Twitter and Reddit. While Twitter and Reddit datasets focus on misinformation detection with political leaning as the sensitive attribute, the Credit dataset addresses payment default detection where age serves as the sensitive attribute. This diversity in application domains demonstrates the broad applicability of our proposed method. The statistics of Credit are as follows:
>
> | Dataset | # Nodes | # Edges | # Attributes | $\gamma_G$ | $\gamma_A$ | Sensitive Attributes | Anomaly Labels |
> | --- | --- | --- | --- | --- | --- | --- | --- |
> | Credit | 30,000 | 1,436,858 | 13 | 0.0983 | 0.2840 | Age | payment default |
>
> The comparison results of DEFEND against all baseline methods on Credit are shown in [Table](https://i.postimg.cc/5yHDgt2C/image.png). The utility-fairness trade-off curves are shown in [Figure](https://i.postimg.cc/jq9p5yMQ/image.png). We can observe that DEFEND maintains its superior performance over all baseline methods on Credit. The accuracy-fairness trade-off curves of DEFEND consistently occupy the upper-left region of the plot, indicating better performance in both accuracy and fairness metrics.
>
> **Visualizations of disentangled representations.**
>
> We provide visualizations of sensitive-irrelevant representations Z_x and sensitive-relevant representations Z_s in [this figure](https://i.postimg.cc/htR1ZGGq/image.png). In the left figure, Z_x is dimensionally reduced by t-SNE and colored by sensitive attribute. We can observe that the points with different sensitive attributes (s=0 and s=1) are well-mixed together, indicating that Z_x successfully maintains sensitive-irrelevant information. In the right figure, we fit Gaussian distributions to Z_s across different sensitive attribute groups. We can observe that while there is considerable overlap, the distributions of Z_s show different central tendencies for s=0 and s=1, with s=1 group slightly shifted towards more positive values. These visualizations demonstrate that our disentanglement approach achieves somewhat separation between sensitive-relevant and sensitive-irrelevant information, even though the separation is not perfect.

---

> ### Author Response · Authors · 2024-12-31
> **Response to Reviewer 6KZ6 (Part 3/3)**
>
> **Qualitative insights for DEFEND's fairness advantages.**
>
> To demonstrate DEFEND's fairness advantages, we present a comparative analysis of anomaly score distributions between DEFEND and the baseline model DOMINANT on Twitter. The visualization shows that DEFEND exhibits more similar density distributions across different groups compared to DOMINANT. The more balanced distributions suggest that DEFEND treats different groups more fairly when assigning anomaly scores. When applying a threshold to determine anomalies, similar distributions ensure comparable detection rates across groups, thus maintaining group fairness. In contrast, DOMINANT's disparate distributions across groups indicate potential bias in its anomaly detection process, as the same threshold would affect different groups disproportionately.
>
> **Algorithmic Transparency.**
>
> We provide a detailed pseudo code of the algorithm in Appendix C to clarify the implementation details. For complete implementation details, we have also made our code publicly available at https://anonymous.4open.science/r/DEFEND.
>
> **Detailed Discussion of Extensions.**
>
> We have discussed the potential extensions of DEFEND to multiple or continuous sensitive attributes in Section 3.5. However, experimental validation is currently limited by the availability of suitable datasets. We plan to construct benchmark datasets for fair graph anomaly detection that incorporate diverse types of sensitive attributes, which will enable comprehensive evaluation of these extensions. Such datasets will facilitate the validation of DEFEND's adaptability to more complex real-world scenarios where sensitive attributes may not be binary.

---

### Review · Reviewer_qUcw · 2024-12-19

**Summary Of Contributions:**

This paper proposes a fairness-aware graph anomaly detection framework, which is built upon representation disentanglement. The contributions of the submission follow as:

- This work proposes the first fairness-aware strategy that is specifically designed for graph anomaly detection.
- The proposed correlation-based regularizer is straightforward, yet effective.
- Experimental evaluation is thorough demonstrating utility-fairness trade-off clearly.

**Audience:**

Yes

**Claims And Evidence:**

Yes

**Requested Changes:**

Please see the Weaknesses.

**Strengths And Weaknesses:**

_Strengths_:

- This is one of the first works focusing on an important research problem.
- The proposed framework is built upon well-established bias mitigation algorithms, where the proposed regularizer, $ L_{corr} $, helps these existing methods to be used for graph anomaly detection. Thus, the overall algorithm is easy to follow.
- Experimental evaluation is thorough in terms of the selection of baselines, ablation study, and sensitivity analyses.

_Weaknesses_:

- Overall, the justification for not using the reconstruction loss for graph topology is not clear to me. The Authors suggest that "Given the inherent bias in graph topology, anomaly detection relies solely on node attributes". However, it is also well-known that the node attributes carry sensitive information too. Thus, the justification for this design choice must be improved.

- The ablation study for the use of reconstruction loss for graph topology (DEFEND+S) can be designed better. Similar to the employment of $ L_{corr} $ for nodal feature reconstruction, for graph topology reconstruction, the effect of sensitive attributes on the reconstruction quality can be reduced by working on a debiased version of the graph topology.

- The overall algorithm design is built upon the assumption that $Z_{x}$ and $Z_{s}$ are independent. Providing some analytical results on real-world data for this assumption can improve the submission.

- Experimental evaluation can be improved by including scalability analyses.

---

> ### Author Response · Authors · 2024-12-31
> **Response to Reviewer qUcw**
>
> **Justification for excluding graph topology reconstruction in anomaly detection.**
>
> Graph topology are inherently more complex, making bias mitigation in structural reconstruction error through correlation constraints less effective, as demonstrated in our experiments with DEFEND+S. When we attempted to constrain topology-based bias using $\mathcal{L}_{corr}$, we observed significant degradation in detection accuracy without proportional fairness gains. While node attributes may contain sensitive information, they can be effectively constrained by correlation constraints. This empirically-driven design choice represents a balance between fairness and utility in anomaly detection. However, we acknowledge that developing specialized approaches for fair topology reconstruction remains an important direction for future work.
>
> **Enhance DEFEND+S with correlation constraints.**
>
> In DEFEND, the anomaly score $o_i$ is evaluated by reconstructing node attributes, and $\mathcal{L} _ {corr}$ constraints the correlation between the anomaly score and predicted sensitive attributes. DEFEND+S is a variant of DEFEND that introduces a dot product decoder to reconstruct graph topology in anomaly detection phase. The anomaly score thus combines both node attribute and graph topology reconstruction errors. This means $\mathcal{L} _ {corr}$ inherently constraints biases  in both node attributes and graph topology. Considering the potential effect of the weight of $\mathcal{L}_{corr}$, we conducted additional experiments with varying $\beta  \in \lbrace1e-15, 5e-15, 1e-12, 5e-12, 1e-10, 5e-10, 1e-9, 5e-9 \rbrace$. The results can be found in [here](https://i.postimg.cc/43Zwy3C1/image.png).
>
> Our experiments show that with structure reconstruction, stronger correlation constraints substantially decrease accuracy while yielding limited fairness improvements. This can be attributed to the distinct characteristics of attribute and structure reconstruction. For attribute reconstruction, $\mathcal{L} _ {corr}$ effectively constrains node-level bias, which aligns with local attribute learning. In contrast, structure reconstruction involves complex patterns across the graph topology, making it more difficult to regulate through correlation constraints. Furthermore, strict constraints on structure reconstruction may lead to the loss of information critical for anomaly detection. These findings indicate that addressing bias in structure reconstruction requires a specialized approach, otherwise using attribute reconstruction error alone as anomaly score might be more beneficial.
>
> **Visualizations of disentangled representations.**
>
> We provide visualizations of sensitive-irrelevant representations Z_x and sensitive-relevant representations Z_s in [this figure](https://i.postimg.cc/htR1ZGGq/image.png). In the left figure, Z_x is dimensionally reduced by t-SNE and colored by sensitive attribute. We can observe that the points with different sensitive attributes (s=0 and s=1) are well-mixed together, indicating that Z_x successfully maintains sensitive-irrelevant information. In the right figure, we fit Gaussian distributions to Z_s across different sensitive attribute groups. We can observe that while there is considerable overlap, the distributions of Z_s show different central tendencies for s=0 and s=1, with s=1 group slightly shifted towards more positive values. These visualizations demonstrate that our disentanglement approach achieves somewhat separation between sensitive-relevant and sensitive-irrelevant information, even though the separation is not perfect.
>
> **Scalability analyses.**
>
> We conducted comprehensive scalability analyses from both theoretical and empirical perspectives to demonstrate DEFEND's computational feasibility. For theoretical analysis, DEFEND achieves time complexity of $O(Ed + Nd)$ and space complexity of $O(Nd + E)$ through our sparse implementation, where $N$ is the number of nodes, $E$ is the number of edges, and $d$ is the attribute dimension. The detailed computational analysis is provided in Appendix D. Empirically, we conducted runtime and peak GPU memory utilization analyses comparing DEFEND with DOMINANT and its variants that require no preprocessing for graph debiasing. We measured the averaged epoch time across 10 runs for each method. Results demonstrate that DEFEND achieves superior performance in both runtime and memory utilization than DOMINANT and its variants.
> | Method | Model | Reddit |  | Twitter |  | Credit |  |
> |--------|--------|---------|---------|-----------|---------|---------|---------|
> | | | Memory (MB) | Runtime (s) | Memory (MB) | Runtime (s) | Memory (MB) | Runtime (s) |
> |-|DOMINANT|8848|0.123|58154|2.932|22974|0.555|
> |FairOD|DOMINANT|8848|0.242|58156|5.945|22976|1.131|
> |HIN|DOMINANT|8848|0.247|58156|6.006|22976|1.136|
> |Corr.|DOMINANT|8848|0.122|58154|3.004|22974|0.558|
> |Ours|DEFEND|7974|0.074|49570|0.176|19984|0.08|

---

### Author Response · Authors · 2024-12-31
**Updated paper**

We appreciate the insightful comments from all reviewers. We have carefully addressed their concerns and made substantial improvements to our manuscript. Here is a summary of our revisions:

1. Revise the introduction to better clarify our motivation and improve framework figure to better illustrate the architecture of DEFEND.
2. Add theoretical analysis of the disentanglement mechanism and provide visualizations to illustrate the disentanglement.
3. Justify our focus on node attributes for anomaly detection and explain challenges with structural reconstruction through additional experiments.
4. Add a new dataset (i.e., Credit) with different anomaly labels and sensitive attributes to show the broader applicability.
5. Provide detailed time and space computational analysis, provide a sparse implementation of DEFEND to enhance its scalability, and conduct experiments to analysis the scalability of DEFEND.
6. Enhance hyperparameter analysis with deeper insights.
7. Add discussion of current limitations and future work.

---

### Decision · Action_Editor_tU5e · 2025-01-31

**Recommendation:** Accept as is

**Comment:**

This paper proposed a novel framework to enhance the fairness in unsupervised graph anomaly detection. Reviewers recognized the technical contributions of this work and also provided many suggestions regarding technical details, experiments, paper writing, etc. During the revision and discussion stage, the authors have provided very detailed responses as well as additional experimental results, which have been incorporated into the revised paper. Reviewers found that their previous concerns have been sufficiently addressed.

**Audience:**

Graph anomaly detection is an important research topic with many real-world applications. TMLR's audience, especially in domains like finance and health, would be very interested in learning the findings of this paper.

**Claims And Evidence:**

This paper aims to enhance the fairness in unsupervised graph anomaly detection. In particular, the authors proposed a new DisEntangle-based FairnEss-aware aNomaly Detection framework on the attributed graph, named DEFEND. Reviewers all agreed that this paper studies an important problem, and the proposed framework is well motivated and clearly explained. Comprehensive evaluations are provided in the paper, which demonstrated the effectiveness of DEFEND against baselines. Overall, the paper is well written, and the claims in the paper have been sufficiently validated.